# ON EVALUATION METRICS
# FOR GRAPH GENERATIVE MODELS

**Rylee Thompson**[1,2], **Boris Knyazev**[1,2,3], **Elahe Ghalebi**[2], **Jungtaek Kim**[4], **Graham W. Taylor**[1,2]
[1] University of Guelph, [2] Vector Institute, [3] Samsung, SAIT AI Lab, Montreal, [4] POSTECH
`{rylee, bknyazev, gwtaylor}@uoguelph.ca`
`elahe.ghalebi@vectorinstitute.ai, jtkim@postech.ac.kr`

## ABSTRACT

In image generation, generative models can be evaluated naturally by visually inspecting model outputs. However, this is not always the case for graph generative models (GGMs), making their evaluation challenging. Currently, the standard process for evaluating GGMs suffers from three critical limitations: i) it does not produce a single score which makes model selection challenging, ii) in many cases it fails to consider underlying edge and node features, and iii) it is prohibitively slow to perform. In this work, we mitigate these issues by searching for *scalar, domain-agnostic, and scalable metrics* for evaluating and ranking GGMs. To this end, we study existing GGM metrics and neural-network-based metrics emerging from generative models of images that use embeddings extracted from a task-specific network. Motivated by the power of Graph Neural Networks (GNNs) to extract meaningful graph representations *without any training*, we introduce several metrics based on the features extracted by an untrained random GNN. We design experiments to thoroughly test and objectively score metrics on their ability to measure the diversity and fidelity of generated graphs, as well as their sample and computational efficiency. Depending on the quantity of samples, we recommend one of two metrics from our collection of random-GNN-based metrics. We show these two metrics to be more expressive than pre-existing and alternative random-GNN-based metrics using our objective scoring. While we focus on applying these metrics to GGM evaluation, in practice this enables the ability to easily compute the dissimilarity between any two sets of graphs *regardless of domain*. Our code is released at: https://github.com/uoguelph-mlrg/GGM-metrics.

## 1 INTRODUCTION

Graph generation is a key problem in a wide range of domains such as molecule generation (Samanta et al., 2020; Popova et al., 2019; Li et al., 2018; Kong et al., 2021; Jin et al., 2020) and structure generation (Bapst et al., 2019; Thompson et al., 2020). An evaluation metric that is capable of accurately measuring the distance between a set of generated and reference graphs is critical for advancing research on graph generative models (GGMs). This is frequently done by comparing empirical distributions of graph statistics such as orbit counts, degree coefficients, and clustering coefficients through Maximum Mean Discrepancy (MMD) (You et al., 2018; Gretton et al., 2006). While these metrics are capable of making a meaningful comparison between generated and real graphs (You et al., 2018), this evaluation method yields a metric for each individual statistic. In addition, recent works have further increased the number of metrics by performing MMD directly with node and edge feature distributions (Goyal et al., 2020), or on alternative graph statistics such as graph spectra (Liao et al., 2019). While this is not an issue provided there is a primary statistic of interest, all metrics are frequently displayed together to approximate generation quality and evaluate GGMs (You et al., 2018; Liao et al., 2019). This process makes it challenging to measure progress as the ranking of generative models may vary between metrics. In addition, the computation of the metrics from You et al. (2018) can be prohibitively slow (Liao et al., 2019; O'Bray et al., 2022), and they are based only on graph structure, meaning they do not incorporate edge and node features. Therefore, they are less applicable in specific domains such as molecule generation where such features are essential. This particular limitation has led to the use of the Neighborhood Subgraph

Pairwise Distance kernel (NSPDK) (Costa & Grave, 2010) in GGM evaluation (Goyal et al., 2020; Podda & Bacciu, 2021; Kawai et al., 2019) as it naturally incorporates edge and node features. However, this metric is still unable to incorporate *continuous* features in evaluation (Costa & Grave, 2010). Faced with a wide array of metrics and ambiguity regarding when each should be the focus, the community needs robust and scalable *standalone* metrics that can consistently rank GGMs.

While less popular, metrics from image generation literature have been successfully utilized in GGM evaluation. These metrics rely on the use of a task-specific neural network to extract meaningful representations of samples, enabling a more straightforward comparison between generated and reference distributions (Preuer et al., 2018; Liu et al., 2019; Thompson et al., 2020). Although these metrics have been validated empirically in the image domain, they are not universally applicable to GGMs. For example, Fréchet Chemnet Distance (Preuer et al., 2018) uses a language model trained on SMILES strings, rendering it unusable for evaluation of GGMs in other domains. Furthermore, a pretrained GNN cannot be applied to datasets with a different number of edge or node labels. Pretraining a GNN for every dataset can be prohibitive, making the use of such metrics in GGM evaluation less appealing than in the more established and standardized image domain.

In image generation evaluation, classifiers trained on ImageNet (Deng et al., 2009) are frequently used to extract image embeddings (Bińkowski et al., 2018; Heusel et al., 2017; Kynkäänniemi et al., 2019; Xu et al., 2018; Naeem et al., 2020). While classifiers such as Inception v3 (Szegedy et al., 2016) are consistently used, recent works have investigated the use of randomly-initialized CNNs with no further training (hereafter referred to as *a random network*) in generative model evaluation. Xu et al. (2018); Naeem et al. (2020) found that a random CNN performs similarly to ImageNet classifiers on natural images and is superior outside of the natural image domain. In the graph domain, random GNNs have been shown to extract meaningful features to solve downstream graph tasks without training (Kipf & Welling, 2017; Morris et al., 2019; Xu et al., 2019). However, the applicability of random GNNs for the evaluation of GGMs remains unexplored.

In this work, we aim to identify one or more scalar metrics that accurately measures the dissimilarity between two sets of graphs to simplify the ranking of GGMs regardless of domain. We tackle this problem by exploring the use of random GNNs in the evaluation of GGMs using metrics that were developed in the image domain. In addition, we perform objective evaluation of a large number of possible evaluation metrics. We design experiments to thoroughly test each metric on its ability to measure the diversity and fidelity (realism) of generated graphs, as well as their sample and computational efficiency. We study three families of metrics: existing GGM evaluation metrics based on graph statistics and graph kernels, which we call *classical metrics*; image domain metrics using a random GNN; and image domain metrics using a pretrained GNN. We aim to answer the following questions empirically: *(Q1) What are the strengths and limitations of each metric? (Q2) Is pretraining a GNN necessary to accurately evaluate GGMs with image domain metrics? (Q3) Is there a strong scalar and domain-agnostic metric for evaluating and ranking GGMs?* Addressing these questions enabled us to reveal several surprising findings that have implications for GGM evaluation in practice. For example, regarding Q1, we identify a failure mode in the classical metrics in that they are poor at measuring the diversity of generated graphs. Consequently, we find several metrics that are more expressive. In terms of Q2, we determine that pretraining is unnecessary to utilize neural-network-based (NN-based) metrics. Regarding Q3, we find two scalar metrics that are appropriate for evaluating and ranking GGMs in certain scenarios; they are scalable, powerful, and can easily incorporate *continuous or discrete* node and edge features. These findings enable computationally inexpensive and domain-agnostic GGM evaluation.

## 2 BACKGROUND & RELATED WORK

Evaluating generative models in any domain is a notoriously difficult task (Theis et al., 2016). Previous work on generative models has typically relied on two families of evaluation metrics: sample-based (Heusel et al., 2017) and likelihood-based (Theis et al., 2016). However, comparing the log-likelihood of autoregressive GGMs is intractable as it requires marginalizing over all possible node orderings (Chen et al., 2021). While recent work learns an optimal ordering and estimates this marginal Chen et al. (2021), it has been shown previously that likelihood may not be indicative of generation quality (Theis et al., 2016). Sample-based evaluation metrics estimate the distance $\rho$ between real and generated distributions $P_r$ and $P_g$ by drawing random samples (Heusel et al., 2017; You et al., 2018; Bińkowski et al., 2018). That is, they compute $\hat{\rho}(\mathbb{S}_g, \mathbb{S}_r) \approx \rho(P_g, P_r)$, with $\mathbb{S}_r = \{\mathbf{x}_1^r, \ldots, \mathbf{x}_m^r\} \sim P_r$ and $\mathbb{S}_g = \{\mathbf{x}_1^g, \ldots, \mathbf{x}_n^g\} \sim P_g$, where $\mathbf{x}_i$ is defined as some feature

vector extracted from a corresponding graph $G_i$. We use sample-based metrics throughout as they are model agnostic and therefore applicable to all GGMs.

## 2.1 CLASSICAL METRICS

Metrics based on graph statistics (You et al., 2018) are standard in evaluating GGMs (Liao et al., 2019; Dai et al., 2020). These metrics set $\mathbf{x}_i$ to be the clustering coefficient, node degree, or 4-node orbit count histograms[1] that are then used to compute the empirical MMD between generated and reference sets $\mathbb{S}_g, \mathbb{S}_r$ (Gretton et al., 2006):

$$\text{MMD}(\mathbb{S}_g, \mathbb{S}_r) := \frac{1}{m^2} \sum_{i,j=1}^{m} k(\mathbf{x}_i^r, \mathbf{x}_j^r) + \frac{1}{n^2} \sum_{i,j=1}^{n} k(\mathbf{x}_i^g, \mathbf{x}_j^g) - \frac{2}{nm} \sum_{i=1}^{n} \sum_{j=1}^{m} k(\mathbf{x}_i^g, \mathbf{x}_j^r), \quad (1)$$

where $k(\cdot, \cdot)$ is a general kernel function. You et al. (2018) proposed a form of the RBF kernel:

$$k(\mathbf{x}_i, \mathbf{x}_j) = \exp\left(-d(\mathbf{x}_i, \mathbf{x}_j)/2\sigma^2\right), \quad (2)$$

where $d(\cdot, \cdot)$ computes pairwise distance, and in that work was chosen to be the Earth Mover's Distance (EMD). This yields three metrics, one for each graph statistic. The computational cost of these metrics may be decreased by using the total variation distance as $d(\cdot, \cdot)$ in Equation 1 (Liao et al., 2019). However, this change leads to an indefinite kernel and undefined behaviour (O'Bray et al., 2022). Therefore, we only compute these metrics using EMD (You et al., 2018). In addition, several works (Goyal et al., 2020; Podda & Bacciu, 2021; Kawai et al., 2019) evaluate GGMs by replacing $k(\cdot, \cdot)$ with the Neighborhood Subgraph Pairwise Distance graph kernel (NSPDK). This metric has the benefit of incorporating *discrete* edge and node features along with the underlying graph structure in evaluation. Similar to the metrics proposed by You et al. (2018), Moreno et al. (2018) extract graph structure properties such as node degree, clustering coefficient, and geodesic distance. However, these properties are then combined into a scalar metric through the Kolmorogov-Smirnov (KS) multidimensional distance (Justel et al., 1997). We exclude KS from our experiments as it is unable to incorporate edge and node features, which is one of the key properties we seek. Finally, note that other domain-specific metrics such as "percentage of valid graphs" exist. Our goal is not to incorporate, eliminate, or evaluate such metrics; they are *properties* of generated graphs, and unlike the metrics described above *do not* provide a comparison to a reference distribution. We believe that such metrics can still provide valuable information in GGM evaluation.

## 2.2 GRAPH NEURAL NETWORKS

We denote a graph as $G = (\mathbb{V}, E)$ with vertices $\mathbb{V}$ and edges $E = \{(i, j) \mid i, j \in \{1, \ldots, |\mathbb{V}|\}\}$. GNNs allow the extraction of a fixed size representation $\mathbf{x}_i$ of an arbitrary graph $G_i$. While many GNN formulations exist (Wu et al., 2020), we consider Graph Isomorphism Networks (GINs) (Xu et al., 2019) as a common GNN. GINs consist of $L$ propagation layers followed by a graph readout layer to obtain $\mathbf{x}_i$. For node $v \in \mathbb{V}$, the node embeddings $\mathbf{h}_v^{(l)}$ at layer $l \in [1, L]$ are computed as:

$$\mathbf{h}_v^{(l)} = \text{MLP}^{(l)}\left(\mathbf{h}_v^{(l-1)} + f^{(l)}\left(\{\mathbf{h}_u^{(l-1)} : u \in \mathcal{N}(v)\}\right)\right), \quad (3)$$

where $\mathbf{h}_v^{(0)}$ is the input feature of node $v$, $\mathbf{h}_v^{(l)} \in \mathbb{R}^d \ \forall l > 0$ denotes a $d$-dimensional embedding of node $v$ after the $l$-th graph propagation layer; $\mathcal{N}(v)$ denotes the neighbors of node $v$; $\text{MLP}^{(l)}$ is a fully-connected neural network; $f^{(l)}$ is some aggregation function over nodes such as mean, max or sum. A graph readout layer with skip connections aggregates features from all nodes at each layer $l \in [1, L]$ and concatenates them into a single $L \cdot d$ dimensional vector $\mathbf{x}_i$ (Xu et al., 2019):

$$\mathbf{x}_i = \text{CONCAT}\left(\text{READOUT}\left(\{\mathbf{h}_v^{(l)} \mid v \in \mathbb{V}\}\right) \mid l \in 1, 2, ..., L\right), \quad (4)$$

where READOUT is similar to $f^{(l)}$ and is often chosen as the mean, max, or sum operation.

## 2.3 NEURAL-NETWORK-BASED METRICS

NN-based metrics utilize a task-specific network to extract descriptive multidimensional embeddings of the input data. In image generation evaluation, the activations of a hidden layer in Inception

---

[1]While any set of graph statistics can be compared using MMD, these three are the most common and hence are evaluated in this work.

Figure 1: The standard process of evaluating GGMs using NN-based metrics.

v3 (Szegedy et al., 2016) are frequently used as vector representations of the input images (Heusel et al., 2017). These NN-based metrics can be applied to GGM evaluation by replacing Inception v3 with a GNN (Thompson et al., 2020; Liu et al., 2019). This prompts the use of a wide range of evaluation metrics that have been studied extensively in the image domain. Computation of these metrics follow a common setup and differ only in how the distance between two sets of data are determined (Figure 1).

**Fréchet Distance (FD)** (Heusel et al., 2017) is one of the most popular generative metrics. FD approximates the graph embeddings as continuous multivariate Gaussians with sample mean and covariance $\mu, C$. The distance between distributions is computed as an approximate measure of sample quality: $\text{FD}(\mathbb{S}_r, \mathbb{S}_g) = \|\mu_r - \mu_g\|_2^2 + \text{Tr}(C_r + C_g - 2(C_r C_g)^{1/2})$.

**Improved Precision & Recall (P&R)** (Kynkäänniemi et al., 2019) decouples the quality of a generator into two separate values to aid in the detection of *mode collapse* and *mode dropping*. Mode dropping refers to the case wherein modes of $P_r$ are underrepresented by $P_g$, while mode collapse describes a lack of diversity within the modes of $P_g$. Manifolds are constructed by extending a radius from each embedded sample in a set to its $k^{\text{th}}$ nearest neighbour to form a hypersphere, with the union of all hyperspheres representing a manifold. Precision is the percentage of generated samples that fall within the manifold of real samples, while Recall is the percentage of real samples that fall within the manifold of generated samples. The harmonic mean ("F1 PR") of P&R is a scalar metric that can be decomposed into more meaningful values in experiments (Lucic et al., 2018).

**Density & Coverage (D&C)** (Naeem et al., 2020) have recently been introduced as robust alternatives for Precision and Recall, respectively. As opposed to P&R which take the union of all hyperspheres to create a single manifold for each set, D&C operate on each samples hypersphere independently. Density is calculated as the number of real hyperspheres a generated sample falls within on average. Coverage is described as the percentage of real hyperspheres that contain a generated sample. The hyperspheres used are found using the $k^{\text{th}}$ nearest neighbour as in P&R. As with P&R, the harmonic mean ("F1 DC") of D&C can be used to create a scalar metric (Lucic et al., 2018).

**MMD** (Gretton et al., 2006) (Equation 1) can also be used to measure the dissimilarity between graph embedding distributions. The original Kernel Inception Distance (KID) (Bińkowski et al., 2018) proposed a polynomial kernel with MMD, $k(\mathbf{x}_i, \mathbf{x}_j) = (\frac{1}{d}\mathbf{x}_i^\top \mathbf{x}_j + 1)^3$, where $d$ is the embedding dimension. The linear kernel $k(\mathbf{x}_i, \mathbf{x}_j) = \mathbf{x}_i \cdot \mathbf{x}_j$ is another parameter-free kernel used with MMD to evaluate generative models (O'Bray et al., 2022). In addition, the RBF kernel (Equation 2) with $d(\cdot, \cdot)$ as the Euclidean distance is widely used (Xu et al., 2018; Gretton et al., 2006). The choice of $\sigma$ in Equation 2 has a significant impact on the output of the RBF kernel, and methods for finding and selecting an optimal value is an important area of research (Bach et al., 2004; Gretton et al., 2012a;b). A common strategy is to select $\sigma$ as the median pairwise distance between the two comparison sets (Garreau et al., 2018; Gretton et al., 2006; Schölkopf et al., 1998). Another strategy is to evaluate MMD using a small set of $\sigma$ values and select the value that maximizes MMD: $\sigma = \arg\max_{\sigma \in \Sigma} \text{MMD}(\mathbb{S}_g, \mathbb{S}_r; \sigma)$ (Sriperumbudur et al., 2009). We combine these ideas in our $\sigma$ selection process, and more details are available in Appendix A. We refer to these metrics as "KD," "MMD Linear," and "MMD RBF."

### 2.4 BENCHMARKING EVALUATION METRICS

Our work is closely related to other works benchmarking evaluation metrics of generative models (O'Bray et al., 2022; Xu et al., 2018). A consistent method for assessing evaluation metrics is to start with the creation of "reference" and "generated" sets $\mathbb{S}_r$ and $\mathbb{S}_g$ that originate from the same real distribution $P_r$. Then, one gradually increases distortion to the generated set while recording each metric's response to the distortion. In the image domain, Xu et al. (2018) develop multiple experiments to test metrics for key properties. The metrics are then subjectively evaluated for these proper-

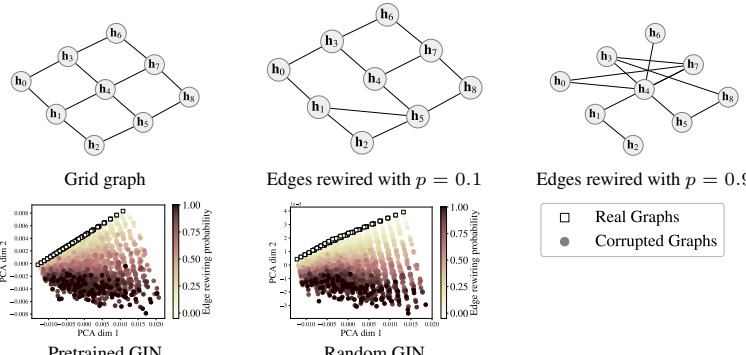

Figure 2: **Top:** Example of a grid graph corrupted by rewiring edges with various probabilities $p$. **Bottom:** Principal component analysis (PCA) of embeddings obtained using the pretrained and random GINs for grid graphs with different rewiring probabilities. For a strong feature extractor, we expect the corrupted graphs to diverge from the real graphs in embedding space as the edge rewiring probability grows. The random GIN extracts as strong a representation as the pretrained GIN which indicates it may be useful in GGM evaluation.

ties by manually analyzing each metric's response across experiments. In concurrent work, O'Bray et al. (2022) introduced additional GGM evaluation metrics. Similar to Xu et al. (2018), the metrics are validated through experiments that slowly increase the level of perturbation in each graph in the generated set. However, O'Bray et al. (2022) incorporated an objective evaluation score by computing the Pearson correlation of each metric with the degree of perturbation. Although objective, evaluation using Pearson correlation is biased in that the relationship between the metric and degree of perturbation need not be linear. Our work is also unique in considering random embeddings.

## 3    THE EFFECTIVENESS OF RANDOM GNNS

While Inception v3 (Szegedy et al., 2016) has found widespread use in evaluation of generated samples (Salimans et al., 2016; Cai et al., 2018; Lunz et al., 2020), there is no analogue in graph-based tasks. This prevents a standardized analysis of GGMs. To tackle this, a single GNN may be pretrained on multiple datasets such that it extracts meaningful embeddings on new datasets (Hu et al., 2020). However, node and edge features often have incompatible dimensions across graph datasets. In addition, the distributions of graphs in the pretraining and target tasks can vary drastically, degrading the performance of a pretrained network. However, random GNNs are capable of extracting meaningful features and solving many graph tasks (Kipf & Welling, 2017; Morris et al., 2019; Xu et al., 2019).[2] Thus, avoiding pretraining and instead utilizing random GNNs may be a viable strategy to bypass these issues. To preliminarily test this approach, we apply permutations to Grid graphs by randomly rewiring edges with probability $p$. Therefore, as we increase $p$, the dissimilarity between the original graphs and permuted graphs increases. Thus, if a GNN is capable of extracting strong representations from graphs, the dissimilarity between graph embeddings should also increase with $p$. We visualize the embeddings extracted from a pretrained GIN (Xu et al., 2019) and a random GIN in Figure 2, and find that they extract very similar representations throughout this experiment. This indicates that both random and pretrained GINs may be capable of evaluating GGMs. We use GIN throughout all experiments due to its theoretical ability to detect graph isomorphism (Xu et al., 2019), however, we also provide a comparison to other common GNNs in Appendix C.6.

## 4    EXPERIMENTS

In this section, we describe key properties of a strong GGM evaluation metric and thoroughly test each metric for these properties. These properties include a metric's ability to correlate with the fidelity and diversity of generated graphs, its sample efficiency, and its computational efficiency. We argue that these properties capture many desired characteristics of a strong evaluation metric and enable reliable ranking of GGMs. In many cases, we adapt the experiment design of Xu et al. (2018) to the graph domain.

---

[2]In the image domain, random CNNs are also beneficial in certain cases, such as large distribution shifts (Naeem et al., 2020).

**Datasets.** We experiment using six diverse graph datasets to test each metric's ability to evaluate GGMs across graph domains (Table 1). In particular, we include common GGM datasets such as Lobster, Grid, Proteins, Community, and Ego (You et al., 2018; Liao et al., 2019; Dai et al., 2020). In addition, we utilize the molecular dataset ZINC (Irwin et al., 2012) strictly to demonstrate the ability of each metric to detect changes in node and edge feature distributions (Section 4.3). We provide thorough descriptions of the datasets in Appendix B.

**GNN feature extractor.** As the GGM literature frequently uses small datasets, the sample efficiency of each metric is extremely important. Furthermore, as the dimensionality of the graph embedding $\mathbf{x}$ is a key factor in many of the metrics in Section 2.3, there is a bias towards minimizing the length of $\mathbf{x}$ while retaining discriminability. As seen in Equation 4, the number of propagation rounds $L$ and the node embedding size $d$

Table 1: A summary of the datasets.

| DATASET | #SAMPLES | $|\mathbb{V}|$ | $|E|$ | NODE/EDGE FEATURES |
|---|---|---|---|---|
| Grid | 100 | 100-400 | 360-1368 | ✗ |
| Lobster | 100 | 10-100 | 10-100 | ✗ |
| Proteins | 918 | 100-500 | 186-1575 | ✗ |
| Ego | 757 | 50-399 | 57-1071 | ✗ |
| Community | 500 | 60-160 | 300-1800 | ✗ |
| ZINC | 1000 | 10-50 | 22-82 | ✓ |

are directly responsible for determining the dimensionality of $\mathbf{x}$. In addition, You et al. (2020) demonstrate that the choice of $L$ is one of the most critical for performance across diverse graph tasks. Thus, in our experiments we consider GIN models (Equations 3 and 4) with $L \in [2, 3, \ldots, 7]$, and $d \in [5, 10, \ldots, 40]$. We randomly select 20 architectures inside these ranges to test in our experiments using both randomly initialized and pretrained GINs. The pretraining process tasks each network with graph classification and the methodology is described in Appendix B. Results for metrics computed using a pretrained GIN in individual experiments are left to Appendix C.1. However, we summarize these results in Table 3 in Section 5 to facilitate discussion. We use node degree features expressed as an integer as an inexpensive way to improve discriminability in both random and pretrained networks. In practice, we utilize orthogonal weight initialization (Saxe et al., 2014) in the random networks as it produces metrics with slightly lower variance across initializations.

**Evaluating the evaluation metrics.** All experiments are designed to begin with $P_g \approx P_r$, and to have a monotonically increasing degree of perturbation $t \in [0, 1]$ that is a measure of the dissimilarity between $\mathbb{S}_g$ and $\mathbb{S}_r$. We evaluate each metric objectively by computing the Spearman rank correlation between the metric scores $\hat{\rho}$ and the degree of perturbation $t$. Spearman rank correlation is preferable to Pearson correlation as it avoids any bias towards a linear relationship. All metrics are normalized such that $\hat{\rho} = 0$ if $P_r = P_g$ meaning that $\hat{\rho}$ should increase with $t$ for a strong metric and a rank correlation of 1.0 is assumed to be ideal. We test each metric and GIN architecture combination across 10 random seeds which affects GIN model weights (if applicable) and perturbations applied. To report the results for a given metric, we first compute the rank correlation for a single random seed (which varies model weights and perturbations applied, if applicable), experiment (e.g. edge rewiring), dataset (e.g. Grid) and GIN configuration (if applicable, e.g. $L = 4, d = 25$). We then aggregate the rank correlation scores across any combination of these factors of variation.

### 4.1 MEASURING FIDELITY

One of the most important properties of a metric is its ability to measure the fidelity of generated samples. To test metrics for fidelity we construct two experiments. The first experiment tests the metric's ability to detect various amounts of random samples mixed with real samples (Xu et al., 2018), while the second experiment slowly decreases the quality of graphs by randomly rewiring edges (O'Bray et al., 2022). Each of these experiments begin with $\mathbb{S}_g$ as a copy of $\mathbb{S}_r$, which is itself a copy of the dataset.

In the first experiment, we utilize random graphs to impact the quality of $\mathbb{S}_g$. To decrease the similarity between $\mathbb{S}_r$ and $\mathbb{S}_g$, we slowly *mix random graphs* by increasing the ratio $t$ of random graphs to real graphs in $\mathbb{S}_g(t)$. We simultaneously remove real graphs such that $|\mathbb{S}_g|$ is constant throughout. The random graphs are Erdős-Rényi (E-R) graphs (Erdős & Rényi, 1960) with sizes and $p$ values chosen to resemble $\mathbb{S}_r$: for every $G_r \in \mathbb{S}_r$, there is a corresponding E-R graph $G_g$ with $|\mathbb{V}|_g = |\mathbb{V}|_r$ and $p = \frac{|E|_r}{|\mathbb{V}|_r^2}$, which is the sparsity of $G_r$.

The second experiment increases distance between $P_r$ and $P_g$ by randomly *rewiring edges* in $\mathbb{S}_g$. Here, the degree of perturbation $t$ is the probability of rewiring each edge $(i, j) \in E$. For each $G \in \mathbb{S}_g$ and each $(i, j) \in E$, a sample $x_{i,j} \sim \text{Bernoulli}(t)$ is drawn. Edges with $x_{i,j} = 1$ are rewired, another sample $y_{i,j} \sim \text{Bernoulli}(0.5)$ is drawn to decide which node $\{i, j\}$ of the edge is kept, and a new connection for this edge is chosen uniformly from $\mathbb{V}$.

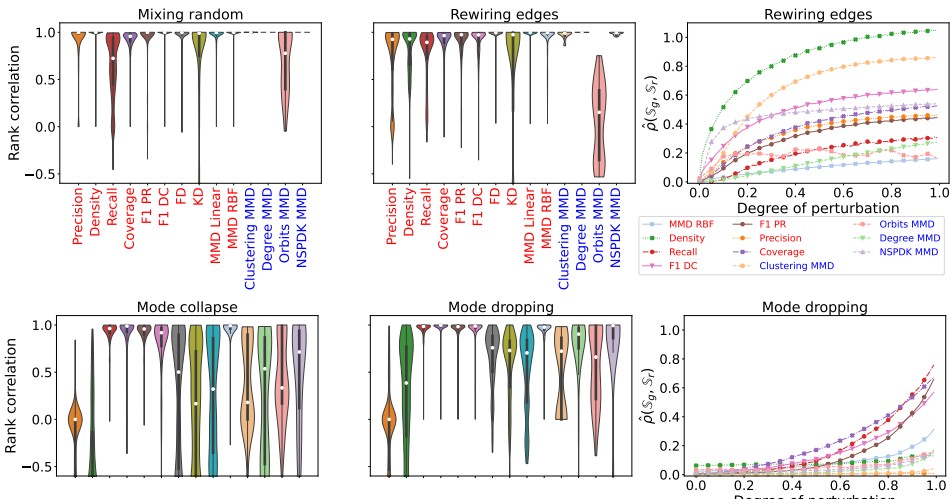

Figure 3: Results from the fidelity (top) and diversity experiments (bottom). NN-based metrics using a random GIN are highlighted in red, while classical metrics are in blue. Results are aggregated across all datasets and all GIN configurations (if applicable). For the violin plots, white dots are the median, thick black lines are the IQR, and thin black lines are the whiskers. The plots on the right show the mean value of select metrics throughout a given experiment. **Fidelity:** Several NN-based and classical metrics perform nearly optimally across both experiments with median rank correlations close to 1.0. **Diversity:** Classical metrics are below average in the mode collapse experiment, and suboptimal in the mode dropping experiment. Scalar metrics such as MMD RBF have median rank correlations close to 1.0 and perform extremely well across both experiments.

**Results.** With the exception of Recall and Orbits MMD, the majority of tested metrics excel in these experiments as indicated by a rank correlation close to 1.0 (Figure 3, top). However, Recall is specifically designed to measure diversity of $\mathbb{S}_g$ rather than fidelity, and its low sensitivity to fidelity here is expected. Surprisingly, Coverage demonstrates strong sensitivity to the fidelity of $\mathbb{S}_g$ although it is also designed to measure the diversity of $\mathbb{S}_g$. In addition, we repeat the mixing experiment with graphs generated by GRAN (Li et al., 2018) and obtain similar results (see Appendix C.4).

## 4.2 MEASURING DIVERSITY

The next property we investigate is a metric's ability to measure the diversity of generated samples in $\mathbb{S}_g$. We focus on two common pitfalls of generative models that a strong metric must be sensitive to: mode dropping and mode collapse. We test each metric for both sensitivities independently by adapting two experiments from Xu et al. (2018) to the graph domain, both of which begin by clustering the dataset using Affinity Propagation (Frey & Dueck, 2007) to identify the modes of $P_r$. Both of these experiments begin with $\mathbb{S}_r$ and $\mathbb{S}_g$ as disjoint halves of the dataset.

To simulate *mode collapse*, we progressively replace each datapoint with its cluster centre. The degree of perturbation $t$ represents the ratio of clusters that have been collapsed in this manner. To simulate *mode dropping*, we progressively remove clusters from $\mathbb{S}_g$. To keep $|\mathbb{S}_g|$ constant, we randomly select samples from the remaining clusters to duplicate. In this experiment, the degree of perturbation $t$ is the ratio of clusters that have been deleted from $\mathbb{S}_g$.

**Results.** In the mode collapse experiment, all classical metrics (You et al., 2018) perform poorly with a rank correlation less than 0.5 (Figure 3, bottom). Classical metrics obtain slightly better, though still suboptimal results in the mode dropping experiment. As expected, Recall and Coverage exhibit strong positive correlation with the diversity of $\mathbb{S}_g$, while Precision and Density are negatively correlated. In addition, several scalar metrics such as MMD RBF and F1 PR exhibit strong correlation with the diversity of $\mathbb{S}_g$ and outperform classical metrics across both experiments.

## 4.3 SENSITIVITY TO NODE AND EDGE FEATURES

In this experiment, we measure the sensitivity of each metric to changes in node or edge feature distributions while the underlying graph structure is static. Similar to the rewiring edges experiment, this is performed by randomizing features with a monotonically increasing probability $t$, and we provide more thorough descriptions of these experiments in Appendix B. We exclude metrics from You

et al. (2018) in these experiments as they are unable to incorporate both edge and node features in evaluation. We find all NN-based metrics and NSPDK MMD are sensitive to these perturbations, and a summary is provided in Table 3.

## 4.4 SAMPLE EFFICIENCY

Small datasets are frequently used in the GGM literature so the notion of *sample efficiency* is important. In this experiment, we determine the sample efficiency of each metric by finding the minimum number of samples to discriminate a set of random graphs $\mathbb{S}_g$ from real samples $\mathbb{S}_r$. The random graphs are E-R graphs generated using the same process described in Section 4.1. We begin this experiment by sampling two disjoint sets $\mathbb{S}'_r$ and $\mathbb{S}''_r$ from $\mathbb{S}_r$, and a set of random graphs $\mathbb{S}'_g$ from $\mathbb{S}_g$ with $|\mathbb{S}'_r| = |\mathbb{S}''_r| = |\mathbb{S}'_g| = n$ and small $n$. A metric with high sample efficiency should measure $\hat{\rho}(\mathbb{S}'_r, \mathbb{S}''_r) < \hat{\rho}(\mathbb{S}'_r, \mathbb{S}'_g)$ with a small $n$. Rather than using rank correlation, in this experiment we record the sample efficiency of each metric as the smallest $n$ where $\hat{\rho}(\mathbb{S}'_r, \mathbb{S}''_r) < \hat{\rho}(\mathbb{S}'_r, \mathbb{S}'_g) \ \ \forall i \geq n$. All of the metrics based on $K$-nearest neighbours and many of the classical metrics exhibit high sample efficiency and require a minimal number of samples to correctly score $\mathbb{S}''_r$ and $\mathbb{S}'_g$ (Table 3).

## 4.5 COMPUTATIONAL EFFICIENCY

Computational efficiency is the final property of an evaluation metric that we examine. Metrics that are efficient to compute are ideal as they can be easily used throughout training to measure progress and perform model selection. Graph datasets can scale in several dimensions: the number of samples, average number of nodes, and average number of edges. We make use of E-R graphs to generate graphs with an arbitrary number of nodes and edges enabling us to independently scale graphs in each dimension. The results highlighting each metric's computational efficiency as dataset size increases to 10,000 samples is shown in Table 3, while the results for all dimensions are presented in Figure 10 in Appendix C. The classical metrics (You et al., 2018) quickly become prohibitive to compute as the number of samples increase, while the NN-based metrics are faster by several orders of magnitude and are inexpensive to compute at any scale.

## 4.6 GGM SELECTION

While NN-based metrics such as MMD RBF and F1 PR have performed consistently in our experiments, this does not necessarily mean they are suited for evaluating GGMs. With the lack of an Inception v3 (Szegedy et al., 2016) analogue for GGM evaluation, these metrics must be consistent across model parameterizations. Thus, in this section we evaluate two popular generative models, GRAN (Liao et al., 2019) and GraphRNN (You et al., 2018) on the Grid dataset. To measure the variance induced by changing model parameterizations, we compute $\hat{\rho}(\mathbb{S}_g, \mathbb{S}_r)$ across 10 different random GINs. We use the strongest GIN configuration along with a strong NN-based metric, MMD RBF, both of which we identify in Section 5. We simulate the model selection process by evaluating models at various stages of training and find that metrics from You et al. (2018) are unable to unanimously rank GGMs (Table 2). As GGMs are frequently evaluated by considering *all* of these metrics together, this highlights the difficulty that may arise during model selection with this evaluation process. Furthermore, we find that MMD RBF is extremely low variance across random GINs indicating its promise for evaluating GGMs. Additional strong NN-based metrics such as F1 PR and F1 DC are tested in Table 12 in Appendix C.8 and are also found to be low variance. Note that we also compute all metrics using a 50/50 split of the dataset. This provides information regarding what score two sets of indistinguishable graphs may receive and represents an ideal value for the given dataset. We suggest that future work also follows this process as it provides a sense of scale and improves interpretability of the results.

## 5 DISCUSSION

We aggregate the results across experiments for a high-level overview of each metric's strengths and weaknesses in Table 3. Metrics computed using a pretrained GIN are nearly indistinguishable from those using a random GIN across rank correlation experiments (columns 3–4). Although metrics using a pretrained GIN benefit from slightly higher sample efficiency, they can have higher variance across model parameters than a simple random initialization (Table 12 in Appendix C.8). Similar to

Table 3: Summary of each metric's performance across experiments. Column headings indicate the experiment across which results are aggregated. NN-based metrics are aggregated across all GIN configurations using random networks unless otherwise stated. Computational efficiency is taken to be the maximum recorded time reported in Figure 10. Values reported are the mean ± std. error, and the average in the final row is taken strictly across the NN-based metrics. Cells are colored if the results can be interpreted objectively for the given experiment (i.e., experiments that use rank correlation to measure performance).

| METRIC | FIDELITY | DIVERSITY | FIDELITY & DIVERSITY (RANDOM/PRETRAINED) | | NODE/EDGE FEATS. | SAMPLE EFF. (RANDOM/PRETRAINED) | | COMP. EFF. (S) |
|---|---|---|---|---|---|---|---|---|
| Orbits MMD | $0.37 \pm 0.048$ | $0.49 \pm 0.046$ | $0.43 \pm 0.034$ | | N/A | $122 \pm 22$ | | $1.4e^4$ |
| Degree MMD | $1.00 \pm 0.000$ | $0.51 \pm 0.061$ | $0.76 \pm 0.035$ | | N/A | $9 \pm 1$ | | $7.5e^3$ |
| Clustering MMD | $0.99 \pm 0.003$ | $0.43 \pm 0.047$ | $0.72 \pm 0.030$ | | N/A | $7 \pm 0$ | | $1.1e^5$ |
| NSPDK MMD | $0.99 \pm 0.001$ | $0.78 \pm 0.050$ | $0.88 \pm 0.028$ | | $1.00 \pm 0.000$ | $8 \pm 1$ | | $382$ |
| FD | $0.98 \pm 0.002$ | $0.44 \pm 0.013$ | $0.71 \pm 0.008$ | $0.74 \pm 0.007$ | $0.96 \pm 0.010$ | $58 \pm 3$ | $55 \pm 3$ | $4.5$ |
| KD | $0.62 \pm 0.015$ | $0.32 \pm 0.014$ | $0.47 \pm 0.010$ | $0.58 \pm 0.009$ | $0.94 \pm 0.011$ | $89 \pm 4$ | $63 \pm 3$ | $5.1$ |
| Precision | $0.82 \pm 0.007$ | $-0.25 \pm 0.010$ | $0.29 \pm 0.011$ | $0.29 \pm 0.011$ | $0.99 \pm 0.001$ | $7 \pm 0$ | $7 \pm 0$ | $18$ |
| Recall | $0.70 \pm 0.007$ | $0.93 \pm 0.003$ | $0.82 \pm 0.004$ | $0.81 \pm 0.005$ | $0.80 \pm 0.018$ | $7 \pm 0$ | $7 \pm 0$ | $18$ |
| Density | $0.90 \pm 0.004$ | $-0.10 \pm 0.015$ | $0.40 \pm 0.011$ | $0.36 \pm 0.012$ | $0.99 \pm 0.001$ | $7 \pm 0$ | $7 \pm 0$ | $12$ |
| Coverage | $0.91 \pm 0.003$ | $0.95 \pm 0.003$ | $0.93 \pm 0.002$ | $0.93 \pm 0.003$ | $0.99 \pm 0.000$ | $7 \pm 0$ | $7 \pm 0$ | $12$ |
| F1 PR | $0.92 \pm 0.004$ | $0.93 \pm 0.003$ | $0.93 \pm 0.003$ | $0.93 \pm 0.002$ | $0.99 \pm 0.000$ | $7 \pm 0$ | $7 \pm 0$ | $18$ |
| F1 DC | $0.95 \pm 0.002$ | $0.86 \pm 0.007$ | $0.91 \pm 0.004$ | $0.88 \pm 0.004$ | $0.99 \pm 0.000$ | $7 \pm 0$ | $7 \pm 0$ | $12$ |
| MMD Linear | $0.98 \pm 0.002$ | $0.37 \pm 0.012$ | $0.68 \pm 0.008$ | $0.75 \pm 0.007$ | $0.99 \pm 0.005$ | $57 \pm 3$ | $31 \pm 2$ | $4.5$ |
| MMD RBF | $0.97 \pm 0.002$ | $0.95 \pm 0.003$ | $0.96 \pm 0.002$ | $0.97 \pm 0.002$ | $1.00 \pm 0.001$ | $42 \pm 2$ | $12 \pm 1$ | $120$ |
| Average (NN-based) | $0.88 \pm 0.039$ | $0.54 \pm 0.144$ | $0.71 \pm 0.078$ | $0.72 \pm 0.076$ | $0.96 \pm 0.019$ | $29 \pm 10$ | $20 \pm 7$ | — |

Table 3, we aggregate the results across all experiments and metrics for a specific GIN configuration in Table 4b in Appendix C. We find that the mean rank correlation taken across NN-based metrics and all experiments is approximately 10× higher variance than the mean taken across GIN configurations (0.078 and 0.007, respectively). In general, our findings indicate that the key to strong GIN-based GGM evaluation metrics is *the choice of metric rather than the specific GNN used to extract graph embeddings*.

In practice, to measure the distance between two sets of graphs we recommend the use of either the MMD RBF or F1 PR metrics. Although we found the choice of GIN architecture to be unimportant, we suggest the use of a random GIN with 3 rounds of graph propagation and a node embedding size of 35 for

Table 2: Evaluation of different GGMs at various percentages of total epochs trained on the Grid dataset. Cells are colored according to their rank in a given column.

| GGM | MMD RBF | Clus. | Deg. | Orbit |
|---|---|---|---|---|
| 50/50 split | $0.042$ | $0.0$ | $6.51e^{-5}$ | $0.018$ |
| GraphRNN-100% | $0.184 \pm 5e^{-5}$ | $4.59e^{-8}$ | $0.032$ | $0.252$ |
| GraphRNN-66% | $0.154 \pm 7e^{-5}$ | $1.69e^{-6}$ | $0.015$ | $0.169$ |
| GRAN-100% | $0.063 \pm 0.001$ | $1.24e^{-6}$ | $1.68e^{-4}$ | $0.037$ |
| GRAN-66% | $0.061 \pm 0.001$ | $3.15e^{-6}$ | $3.20e^{-5}$ | $0.047$ |

consistency in future work. This is the strongest GIN configuration across all experiments (Table 4b in Appendix C.2). Both metrics have been shown to be sensitive to changes in both fidelity and diversity while having minimal variance across random initializations. In addition, they are capable of detecting changes in node and edge feature distributions, making them useful for a wide array of datasets.[3] To highlight this, we provide results for individual datasets in Appendix B.1 and find the performance of GNN-based metrics to be consistent across all datasets. While MMD RBF has slightly stronger correlation with the fidelity and diversity of generated graphs, F1 PR has superior sample and computational efficiency. Thus, we suggest the use of F1 PR in cases where there are fewer samples than the measured sample size of MMD RBF (i.e., 42) or MMD RBF is extremely prohibitive to compute, otherwise, we suggest MMD RBF.

## 6 CONCLUSION

We introduced the use of an untrained random GIN in GGM evaluation, inspired by metrics popular in the image domain. We discovered that pre-existing GGM metrics fail to capture the diversity of generated graphs, and find several random GIN-based metrics that are more expressive while being domain-agnostic at a significantly reduced computational cost. An interesting direction for future work is to provide theoretical justification supporting the use of random GINs, as well as the exploration of more sophisticated GNNs to improve sample efficiency.

---

[3]This is not to say that they are the only metrics that should be used to evaluate GGMs. For example, in the molecular domain percent valid or druglikeness can provide valuable information that is not explicitly measured by our metrics. In addition, if the goal is for generated graphs to resemble a reference set based on a *singular* graph statistic, the metrics from You et al. (2018) are still invaluable.

ACKNOWLEDGMENTS

BK and RT received support from NSERC. RT received funding from the Vector Scholarship in Artificial Intelligence. BK received support from the Ontario Graduate Scholarship. GWT acknowledges support from CIFAR and the Canada Foundation for Innovation. Resources used in preparing this research were provided, in part, by the Province of Ontario, the Government of Canada through CIFAR, and companies sponsoring the Vector Institute: `http://www.vectorinstitute.ai/#partners`.

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

# Appendices

# A    ADDITIONAL METRIC DETAILS

For Precision, Recall, Density, and Coverage, we set $k = 5$ for all experiments (Naeem et al., 2020). For the MMD RBF metric, we compute MMD as $\mathrm{MMD}(\mathbb{S}_g, \mathbb{S}_r) = \max\{\mathrm{MMD}(\mathbb{S}_g, \mathbb{S}_r; \sigma) \mid \sigma \in \Sigma\}$, where the values of $\Sigma$ are multiplied by the mean pairwise distance of $\mathbb{S}_g$ and $\mathbb{S}_r$. While the median pairwise distance is the heuristic (Garreau et al., 2018), we find the mean to be more robust in our experiments. Before this scaling factor is applied, we use $\Sigma = \{0.01, 0.1, 0.25, 0.5, 0.75, 1.0, 2.5, 5.0, 7.5, 10.0\}$. For all baseline metrics from You et al. (2018), we use the hyperparameters chosen in their open-source code. For the NSPDK MMD metric we use the open-source code from Goyal et al. (2020)

# B    ADDITIONAL EXPERIMENTAL DETAILS

Following Section 2.2, we denote a graph as $G = (\mathbb{V}, E)$ with vertices $\mathbb{V}$ and edges $E = \{(i, j) \mid i, j \in \{1, \ldots, |\mathbb{V}|\}\}$.

Node features for node $i \in \mathbb{V}$ are denoted as $\mathbf{h}_i \in \mathbb{R}^b$ ($\mathbf{h}_i^{(0)}$ in Section 2.2). Similarly, $\boldsymbol{A} \in \mathbb{R}^{n \times n \times a}$ stores edge information, with the $(i, j)^{th}$ entry corresponding to the edge features for edge $(i, j)$. Unless otherwise specified in Appendix B.1, $\boldsymbol{A}$ is not utilized and we set $\mathbf{h}_i$ to the degree of node $i$ expressed as an integer as an inexpensive way to improve discriminability. We handle edge features by concatenating $\boldsymbol{A}_{i,j,:}$ to messages from node $i$ to node $j$ at each round of graph propagation (Hu et al., 2020).

## B.1    DATASETS

We perform experiments using a variety of datasets with varying sizes and characteristics to thoroughly test each metric's ability to evaluate GGMs across domains. We place a slight emphasis on datasets that are frequently used in the GGM literature.

**Lobster.** A set of 100 stochastic graphs where each node is at most 2 hops away from a backbone path. We generate lobster graphs with $10 \leq |\mathbb{V}| \leq 100$ (Dai et al., 2020).

**Grid.** 100 2D grid graphs generated with $100 \leq |\mathbb{V}| \leq 400$ (Dai et al., 2020; You et al., 2018; Liao et al., 2019).

**Proteins.** 918 protein graphs where nodes are amino acids and two nodes are connected by an edge if they are less than 6 Angstroms away (Dobson & Doig, 2003). We select graphs with $100 \leq |\mathbb{V}| \leq 500$ (You et al., 2018; Liao et al., 2019; Dai et al., 2020).

**Ego.** 757 3-hop ego networks with $50 \leq |\mathbb{V}| \leq 399$ (You et al., 2018). Graphs are extracted from the CiteSeer network where nodes represent documents and edges represent citation relationships (Sen et al., 2008).

**Community.** 500 two-community graphs with $60 \leq |\mathbb{V}| \leq 160$. Each community is generated by the Erdős-Rényi model (E-R) (Erdős & Rényi, 1960) with $n = |\mathbb{V}|/2$ and $p = 0.3$. We then add $0.05|\mathbb{V}|$ inter-community edges with uniform probability (You et al., 2018).

**ZINC.** 250k real-world molecular graphs (Irwin et al., 2012) of which we randomly select 1000 samples for efficiency with $10 \leq |\mathbb{V}| \leq 50$. We set $\mathbf{h}_i$ to be a one-hot encoding of the element of node $i$, and $\boldsymbol{A}_{i,j,:}$ to be a one-hot encoding $\boldsymbol{a} \in \mathbb{R}^a$ of the bond type between nodes $i$ and $j$. This dataset is unique in that it is the only one to naturally contain node and edge features. Thus, we can use this dataset to determine a metric's sensitivity to changes in the edge and node feature distributions without generating artificial labels.

## B.2    MEASURING DIVERSITY

We employ Affinity Propagation (Frey & Dueck, 2007) to automatically determine the number of clusters, and set the similarity between graphs to be the value obtained from the WL-subtree kernel (Shervashidze et al., 2011) using the GraKel Python library (Siglidis et al., 2020). This clustering method avoids any dependency on the GNN model architecture and ensures the clusters found are consistent across all experiments.

### B.3 RANDOMIZING EDGE FEATURES

This experiment is performed exclusively on the ZINC dataset as it naturally contains edge features. Here, the degree of perturbation $t$ is the probability of randomizing each edge feature $\boldsymbol{A}_{i,j,:}, \forall (i,j) \in E$. For each $G \in \mathbb{S}_g$ and each edge $(i,j) \in E$, a sample $x_{i,j} \sim \text{Bernoulli}(t)$ is drawn. Edges with $x_{i,j} = 1$ have their edge feature randomized. As these features are a one-hot encoding, we sample the new edge feature uniformly from the set of valid edge features. The results of this experiment are shown in Figure 9.

### B.4 RANDOMIZING NODE FEATURES

This experiment is performed exclusively on the ZINC dataset as it naturally contains node features. Here, the degree of perturbation $t$ is the probability of randomizing each node feature $\mathbf{h}_i, \forall i \in \mathbb{V}$. For each $G \in \mathbb{S}_g$ and each node $i \in \mathbb{V}$, a sample $x_i \sim \text{Bernoulli}(t)$ is drawn. Nodes with $x_i = 1$ have their feature randomized. As these features are a one-hot encoding, we sample the new node feature uniformly from the set of valid node features. The results of this experiment are shown in Figure 9.

### B.5 COMPUTATIONAL EFFICIENCY

All experiments in this section were conducted on an Intel Platinum 8160F Skylake @ 2.1Ghz with 4 CPU cores. We benchmark NN-based metrics using the most expensive GIN configuration tested in our experiments. This was a configuration with 7 propagation layers and a node embedding size of 20. The results for this experiment on CPU are shown in Figure 10 and on GPU in Figure 11. In addition, the approximate RAM usage for each metric throughout these experiments are shown in Figure 12 and Table 9.

**Scaling number of samples.** We generate various sets of E-R graphs (Erdős & Rényi, 1960) to scale the number of samples in a dataset independently from the number of nodes and number of edges per graph. We generate sets of graphs with the number of samples in $[100, 1000, 2000, \ldots, 10\text{k}]$ with $|\mathbb{V}| = 50$ for all graphs. We choose $p = \frac{|E|_{\text{avg}}}{|\mathbb{V}|_{\text{avg}}^2}$ for all graphs, where the averages are taken across the Proteins dataset.

**Scaling number of edges.** To scale the number of edges independently, we generate 50 E-R graphs per set with $|\mathbb{V}| = 1000$. The graphs are generated with $p$ in $[0.01, 0.1, 0.2, \ldots, 1.0]$.

**Scaling number of nodes.** We generate 50 graphs of a constant size per set with $|\mathbb{V}|$ in $[1000, 10\text{k}, 20\text{k}, \ldots, 100\text{k}]$. For each set, we use $p = \frac{10000}{|\mathbb{V}|^2}$ to generate graphs with approximately 10000 edges per graph.

### B.6 GGM SELECTION

We train GRAN (Liao et al., 2019) and GraphRNN (You et al., 2018) with 80% of the graphs randomly selected for training. We use the implementations in the official GitHub repositories and train using the recommended hyperparameters. We then generate $n$ graphs from each model where $n$ is the size of the dataset, and use the remaining 20% of graphs as $\mathbb{S}_r$.

### B.7 GNN PRETRAINING

We isolate the comparison between randomly initialized and pretrained weights to datasets without node or edge features (i.e. excluding ZINC). The GNN is trained under the multiclass classification setting to classify a given graph into the remaining datasets. To increase the difficulty of the problem, for each graph $G_r$ in the remaining datasets, we generate an E-R graph (Erdős & Rényi, 1960) $G_g$ with $|\mathbb{V}|_g = |\mathbb{V}|_r$ and $p = \frac{|E|_r}{|\mathbb{V}|_r^2}$. These E-R graphs have separate labels, thereby doubling the number of classes present. We use the same optimizer across all experiments and select the model with the lowest validation loss. We train each configuration across 10 random seeds to perform a fair comparison with the 10 different random initializations used in our experiments.

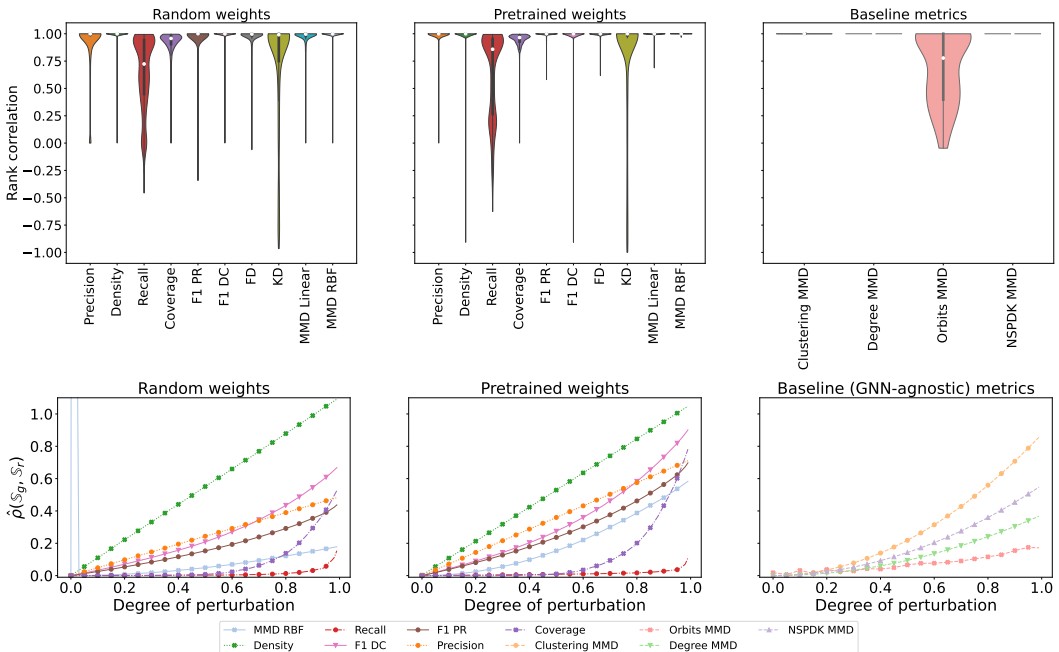

Figure 4: Results from the **mixing random graphs** experiment across all datasets and all GIN configurations (if applicable).

# C  ADDITIONAL RESULTS

Here we provide a variety of supplementary experiments that were not included in the main body due to page limits. Unless otherwise specified, results for all GIN-based metrics utilize a random GIN, are aggregated across all GIN configurations, and all GIN configurations use summation aggregation.

## C.1  INDIVIDUAL EXPERIMENTS

We provide violin plots for each individual experiment in Figures 4 through 8.

## C.2  ALL EXPERIMENTS GROUPED BY GIN CONFIGURATION

We aggregate results by GIN configuration across all experiments in Tables 4. In addition, we aggregate results by GIN configuration and metric combination and highlight the top 20 results in Table 5.

## C.3  ALL RESULTS GROUPED BY DATASET

For a more in-depth comparison of metrics, here we provide results for all experiments grouped by the dataset used. The results are shown in Tables 6 and 7. We find that while the performance of each metric does fluctuate slightly across datasets, all of our conclusions drawn in Section 5 still hold regardless of dataset and there are no obvious outliers. F1 PR and MMD RBF are consistently the highest scoring metrics, and with the exception of the Community dataset all baseline metrics (You et al., 2018) are poor at measuring the diversity of generated graphs. Notably, the performance of Clustering MMD on diversity experiments is affected by the Lobster and Grid datasets; these graphs are triangle-free and thus all nodes have a clustering coefficient of zero. However, even if these datasets are excluded from the analysis, Clustering MMD is still suboptimal on the remainder of the datasets.

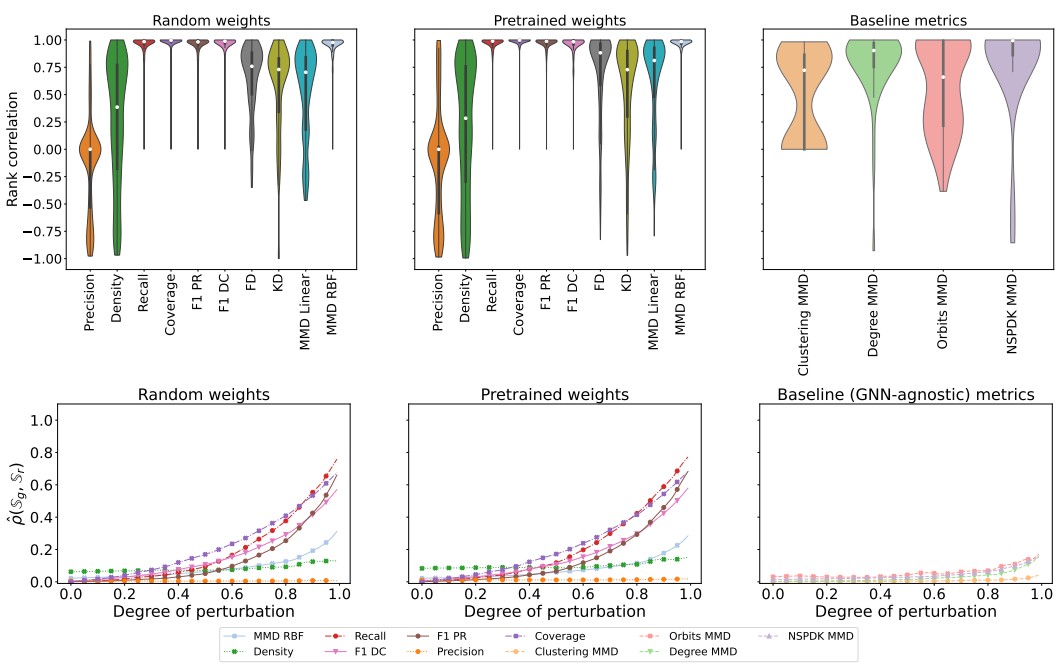

Figure 5: Results from the **rewiring edges** experiment across all datasets and all GIN configurations (if applicable).

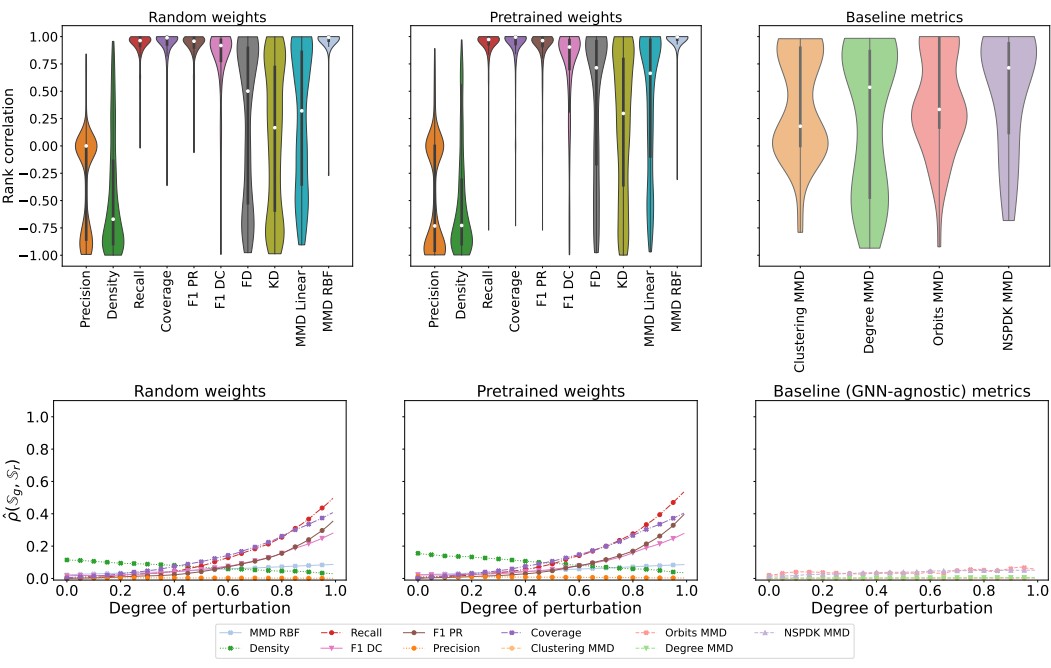

Figure 6: Results from the **mode collapse** experiment across all datasets and all GIN configurations (if applicable).

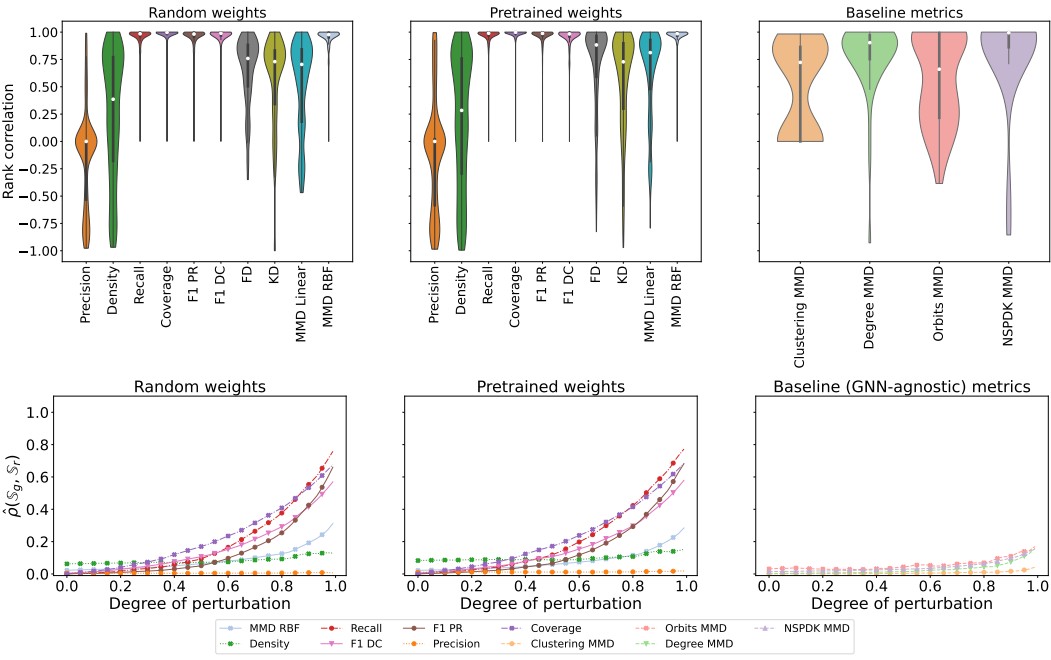

Figure 7: Results from the **mode dropping** experiment across all datasets and all GIN configurations (if applicable).

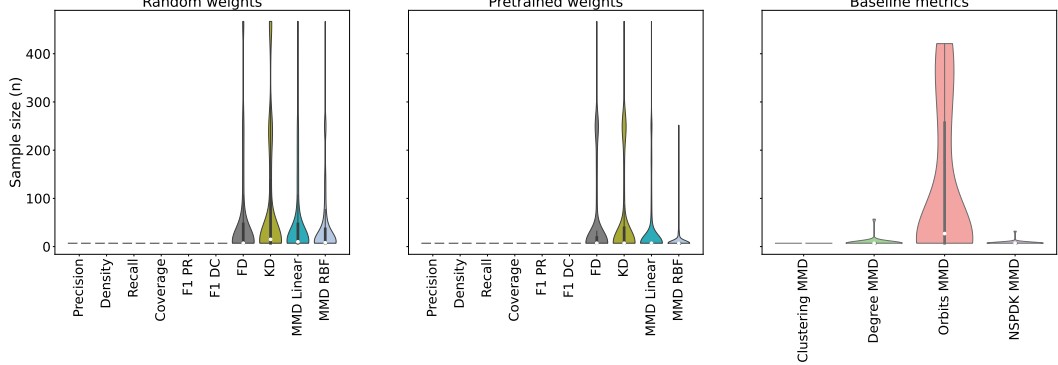

Figure 8: Results from the **sample efficiency** experiment across all datasets and all GIN configurations (if applicable).

Table 4: Results across fidelity, diversity, and sample efficiency experiments and all datasets grouped by GIN configuration. N is the node embedding size, P is the number of graph propagation rounds, and G is the dimensionality of the obtained graph embedding. Configurations are sorted by their mean rank correlation across fidelity and diversity experiments.

(a) Pretrained GIN

| GIN CONFIG. | FIDELITY | DIVERSITY | FIDELITY & DIVERSITY | SAMPLE EFF. |
|---|---|---|---|---|
| N: 25, P: 3, G: 50 | $0.90 \pm 0.007$ | $0.59 \pm 0.019$ | $0.75 \pm 0.011$ | $17 \pm 2$ |
| N: 35, P: 5, G: 140 | $0.89 \pm 0.008$ | $0.59 \pm 0.019$ | $0.74 \pm 0.011$ | $19 \pm 2$ |
| N: 35, P: 3, G: 70 | $0.91 \pm 0.007$ | $0.56 \pm 0.020$ | $0.74 \pm 0.011$ | $19 \pm 2$ |
| N: 30, P: 5, G: 120 | $0.90 \pm 0.008$ | $0.57 \pm 0.019$ | $0.73 \pm 0.011$ | $20 \pm 2$ |
| N: 40, P: 6, G: 200 | $0.89 \pm 0.008$ | $0.57 \pm 0.019$ | $0.73 \pm 0.011$ | $20 \pm 2$ |
| N: 40, P: 2, G: 40 | $0.90 \pm 0.007$ | $0.56 \pm 0.020$ | $0.73 \pm 0.011$ | $22 \pm 3$ |
| N: 40, P: 5, G: 160 | $0.87 \pm 0.009$ | $0.59 \pm 0.019$ | $0.73 \pm 0.011$ | $21 \pm 2$ |
| N: 20, P: 5, G: 80 | $0.90 \pm 0.008$ | $0.56 \pm 0.020$ | $0.73 \pm 0.011$ | $19 \pm 2$ |
| N: 15, P: 6, G: 75 | $0.89 \pm 0.008$ | $0.56 \pm 0.020$ | $0.73 \pm 0.011$ | $20 \pm 2$ |
| N: 15, P: 2, G: 15 | $0.89 \pm 0.008$ | $0.56 \pm 0.019$ | $0.73 \pm 0.011$ | $23 \pm 3$ |
| N: 20, P: 7, G: 120 | $0.90 \pm 0.007$ | $0.55 \pm 0.020$ | $0.73 \pm 0.011$ | $22 \pm 3$ |
| N: 20, P: 3, G: 40 | $0.91 \pm 0.007$ | $0.54 \pm 0.020$ | $0.72 \pm 0.011$ | $18 \pm 2$ |
| N: 10, P: 7, G: 60 | $0.89 \pm 0.009$ | $0.56 \pm 0.020$ | $0.72 \pm 0.011$ | $22 \pm 3$ |
| N: 5, P: 5, G: 20 | $0.90 \pm 0.008$ | $0.54 \pm 0.020$ | $0.72 \pm 0.012$ | $18 \pm 2$ |
| N: 10, P: 3, G: 20 | $0.89 \pm 0.008$ | $0.55 \pm 0.020$ | $0.72 \pm 0.011$ | $16 \pm 2$ |
| N: 15, P: 7, G: 90 | $0.88 \pm 0.009$ | $0.55 \pm 0.020$ | $0.72 \pm 0.011$ | $20 \pm 2$ |
| N: 25, P: 4, G: 75 | $0.88 \pm 0.009$ | $0.56 \pm 0.020$ | $0.72 \pm 0.011$ | $21 \pm 2$ |
| N: 5, P: 7, G: 30 | $0.89 \pm 0.008$ | $0.54 \pm 0.020$ | $0.72 \pm 0.012$ | $22 \pm 3$ |
| N: 10, P: 2, G: 10 | $0.86 \pm 0.010$ | $0.56 \pm 0.020$ | $0.71 \pm 0.011$ | $26 \pm 3$ |
| N: 5, P: 2, G: 5 | $0.86 \pm 0.009$ | $0.51 \pm 0.019$ | $0.69 \pm 0.011$ | $24 \pm 3$ |
| Average | $0.89 \pm 0.003$ | $0.56 \pm 0.004$ | $0.73 \pm 0.003$ | $20 \pm 1$ |

(b) Random GIN.

| GIN CONFIG. | FIDELITY | DIVERSITY | FIDELITY & DIVERSITY | SAMPLE EFF. |
|---|---|---|---|---|
| N: 35, P: 3, G: 70 | $0.92 \pm 0.008$ | $0.55 \pm 0.019$ | $0.73 \pm 0.011$ | $31 \pm 3$ |
| N: 10, P: 3, G: 20 | $0.92 \pm 0.008$ | $0.55 \pm 0.019$ | $0.73 \pm 0.011$ | $34 \pm 4$ |
| N: 25, P: 3, G: 50 | $0.92 \pm 0.008$ | $0.55 \pm 0.019$ | $0.73 \pm 0.011$ | $31 \pm 3$ |
| N: 35, P: 5, G: 140 | $0.91 \pm 0.008$ | $0.55 \pm 0.019$ | $0.73 \pm 0.011$ | $18 \pm 2$ |
| N: 25, P: 4, G: 75 | $0.91 \pm 0.009$ | $0.55 \pm 0.019$ | $0.73 \pm 0.011$ | $26 \pm 3$ |
| N: 20, P: 3, G: 40 | $0.91 \pm 0.008$ | $0.54 \pm 0.020$ | $0.72 \pm 0.011$ | $34 \pm 4$ |
| N: 40, P: 5, G: 160 | $0.90 \pm 0.008$ | $0.54 \pm 0.019$ | $0.72 \pm 0.011$ | $20 \pm 2$ |
| N: 30, P: 5, G: 120 | $0.90 \pm 0.009$ | $0.54 \pm 0.019$ | $0.72 \pm 0.011$ | $22 \pm 2$ |
| N: 15, P: 7, G: 90 | $0.90 \pm 0.008$ | $0.54 \pm 0.019$ | $0.72 \pm 0.011$ | $26 \pm 3$ |
| N: 40, P: 6, G: 200 | $0.90 \pm 0.008$ | $0.53 \pm 0.020$ | $0.72 \pm 0.011$ | $20 \pm 3$ |
| N: 20, P: 5, G: 80 | $0.90 \pm 0.009$ | $0.54 \pm 0.020$ | $0.72 \pm 0.012$ | $30 \pm 4$ |
| N: 15, P: 6, G: 75 | $0.89 \pm 0.009$ | $0.54 \pm 0.019$ | $0.72 \pm 0.011$ | $26 \pm 3$ |
| N: 20, P: 7, G: 120 | $0.90 \pm 0.007$ | $0.53 \pm 0.020$ | $0.72 \pm 0.011$ | $22 \pm 3$ |
| N: 10, P: 7, G: 60 | $0.90 \pm 0.008$ | $0.53 \pm 0.019$ | $0.71 \pm 0.011$ | $25 \pm 3$ |
| N: 5, P: 7, G: 30 | $0.90 \pm 0.009$ | $0.53 \pm 0.020$ | $0.71 \pm 0.012$ | $25 \pm 3$ |
| N: 5, P: 5, G: 20 | $0.87 \pm 0.011$ | $0.55 \pm 0.019$ | $0.71 \pm 0.012$ | $36 \pm 4$ |
| N: 40, P: 2, G: 40 | $0.80 \pm 0.011$ | $0.55 \pm 0.019$ | $0.68 \pm 0.011$ | $32 \pm 3$ |
| N: 15, P: 2, G: 15 | $0.78 \pm 0.012$ | $0.56 \pm 0.019$ | $0.67 \pm 0.011$ | $46 \pm 5$ |
| N: 10, P: 2, G: 10 | $0.75 \pm 0.014$ | $0.52 \pm 0.021$ | $0.64 \pm 0.013$ | $29 \pm 3$ |
| N: 5, P: 2, G: 5 | $0.74 \pm 0.012$ | $0.50 \pm 0.019$ | $0.62 \pm 0.012$ | $44 \pm 4$ |
| Average | $0.88 \pm 0.013$ | $0.54 \pm 0.003$ | $0.71 \pm 0.007$ | $29 \pm 2$ |

Table 5: Results across fidelity, diversity, and sample efficiency experiments and all datasets grouped by GIN configuration and metric used. N is the node embedding size, P is the number of graph propagation rounds, and G is the dimensionality of the obtained graph embedding. Configurations are sorted by their mean rank correlation across fidelity and diversity experiments and the top 20 results are shown.

(a) Pretrained GIN.

| GIN CONFIG., METRIC | FIDELITY | DIVERSITY | FIDELITY & DIVERSITY | SAMPLE EFF. |
|---|---|---|---|---|
| N: 35, P: 3, G: 70, MMD RBF | $0.99 \pm 0.005$ | $0.97 \pm 0.005$ | $0.98 \pm 0.004$ | $9 \pm 1$ |
| N: 25, P: 4, G: 75, MMD RBF | $0.98 \pm 0.005$ | $0.97 \pm 0.007$ | $0.98 \pm 0.004$ | $10 \pm 2$ |
| N: 20, P: 3, G: 40, MMD RBF | $0.99 \pm 0.006$ | $0.97 \pm 0.013$ | $0.98 \pm 0.007$ | $9 \pm 1$ |
| N: 30, P: 5, G: 120, MMD RBF | $0.98 \pm 0.007$ | $0.97 \pm 0.007$ | $0.97 \pm 0.005$ | $11 \pm 3$ |
| N: 20, P: 5, G: 80, MMD RBF | $0.98 \pm 0.007$ | $0.97 \pm 0.010$ | $0.97 \pm 0.006$ | $9 \pm 1$ |
| N: 40, P: 2, G: 40, MMD RBF | $0.99 \pm 0.004$ | $0.96 \pm 0.014$ | $0.97 \pm 0.007$ | $12 \pm 2$ |
| N: 10, P: 7, G: 60, MMD RBF | $0.98 \pm 0.004$ | $0.96 \pm 0.014$ | $0.97 \pm 0.007$ | $12 \pm 2$ |
| N: 10, P: 2, G: 10, MMD RBF | $0.99 \pm 0.004$ | $0.96 \pm 0.014$ | $0.97 \pm 0.007$ | $19 \pm 5$ |
| N: 10, P: 3, G: 20, MMD RBF | $0.98 \pm 0.006$ | $0.96 \pm 0.013$ | $0.97 \pm 0.007$ | $11 \pm 3$ |
| N: 25, P: 3, G: 50, MMD RBF | $0.98 \pm 0.008$ | $0.96 \pm 0.013$ | $0.97 \pm 0.007$ | $10 \pm 2$ |
| N: 15, P: 7, G: 90, MMD RBF | $0.98 \pm 0.006$ | $0.96 \pm 0.013$ | $0.97 \pm 0.007$ | $9 \pm 1$ |
| N: 20, P: 7, G: 120, MMD RBF | $0.98 \pm 0.006$ | $0.96 \pm 0.014$ | $0.97 \pm 0.008$ | $10 \pm 1$ |
| N: 40, P: 6, G: 200, MMD RBF | $0.98 \pm 0.010$ | $0.97 \pm 0.010$ | $0.97 \pm 0.007$ | $9 \pm 1$ |
| N: 5, P: 5, G: 20, MMD RBF | $0.98 \pm 0.011$ | $0.97 \pm 0.012$ | $0.97 \pm 0.008$ | $11 \pm 2$ |
| N: 15, P: 6, G: 75, MMD RBF | $0.98 \pm 0.010$ | $0.96 \pm 0.014$ | $0.97 \pm 0.009$ | $9 \pm 1$ |
| N: 5, P: 7, G: 30, MMD RBF | $0.98 \pm 0.011$ | $0.96 \pm 0.013$ | $0.97 \pm 0.008$ | $14 \pm 5$ |
| N: 15, P: 2, G: 15, MMD RBF | $0.98 \pm 0.007$ | $0.96 \pm 0.014$ | $0.97 \pm 0.008$ | $17 \pm 5$ |
| N: 35, P: 5, G: 140, MMD RBF | $0.96 \pm 0.015$ | $0.97 \pm 0.008$ | $0.97 \pm 0.008$ | $9 \pm 1$ |
| N: 40, P: 5, G: 160, MMD RBF | $0.97 \pm 0.010$ | $0.96 \pm 0.011$ | $0.96 \pm 0.007$ | $9 \pm 1$ |
| N: 15, P: 2, G: 15, F1 PR | $0.94 \pm 0.011$ | $0.95 \pm 0.010$ | $0.95 \pm 0.007$ | $7 \pm 0$ |

(b) Random GIN.

| GIN CONFIG., METRIC | FIDELITY | DIVERSITY | FIDELITY & DIVERSITY | SAMPLE EFF. |
|---|---|---|---|---|
| N: 10, P: 7, G: 60, MMD RBF | $0.99 \pm 0.002$ | $0.95 \pm 0.013$ | $0.97 \pm 0.007$ | $25 \pm 6$ |
| N: 25, P: 4, G: 75, MMD RBF | $0.99 \pm 0.002$ | $0.95 \pm 0.013$ | $0.97 \pm 0.007$ | $30 \pm 7$ |
| N: 25, P: 3, G: 50, MMD RBF | $0.99 \pm 0.002$ | $0.95 \pm 0.014$ | $0.97 \pm 0.007$ | $51 \pm 10$ |
| N: 20, P: 5, G: 80, MMD RBF | $0.99 \pm 0.005$ | $0.95 \pm 0.014$ | $0.97 \pm 0.007$ | $31 \pm 7$ |
| N: 10, P: 3, G: 20, MMD RBF | $0.99 \pm 0.003$ | $0.95 \pm 0.014$ | $0.97 \pm 0.007$ | $56 \pm 11$ |
| N: 35, P: 3, G: 70, MMD RBF | $0.99 \pm 0.003$ | $0.95 \pm 0.014$ | $0.97 \pm 0.007$ | $52 \pm 10$ |
| N: 20, P: 3, G: 40, MMD RBF | $0.99 \pm 0.005$ | $0.95 \pm 0.014$ | $0.97 \pm 0.007$ | $52 \pm 11$ |
| N: 5, P: 5, G: 20, MMD RBF | $0.99 \pm 0.006$ | $0.95 \pm 0.013$ | $0.97 \pm 0.007$ | $41 \pm 9$ |
| N: 5, P: 7, G: 30, MMD RBF | $0.99 \pm 0.007$ | $0.95 \pm 0.013$ | $0.97 \pm 0.008$ | $25 \pm 7$ |
| N: 30, P: 5, G: 120, MMD RBF | $0.99 \pm 0.008$ | $0.95 \pm 0.014$ | $0.97 \pm 0.008$ | $29 \pm 6$ |
| N: 15, P: 7, G: 90, MMD RBF | $0.98 \pm 0.008$ | $0.96 \pm 0.013$ | $0.97 \pm 0.008$ | $20 \pm 4$ |
| N: 15, P: 6, G: 75, MMD RBF | $0.98 \pm 0.008$ | $0.96 \pm 0.013$ | $0.97 \pm 0.008$ | $21 \pm 4$ |
| N: 20, P: 7, G: 120, MMD RBF | $0.98 \pm 0.007$ | $0.96 \pm 0.013$ | $0.97 \pm 0.008$ | $23 \pm 5$ |
| N: 35, P: 5, G: 140, MMD RBF | $0.98 \pm 0.008$ | $0.95 \pm 0.013$ | $0.97 \pm 0.008$ | $17 \pm 3$ |
| N: 40, P: 6, G: 200, MMD RBF | $0.97 \pm 0.009$ | $0.95 \pm 0.013$ | $0.96 \pm 0.008$ | $17 \pm 4$ |
| N: 40, P: 5, G: 160, MMD RBF | $0.97 \pm 0.010$ | $0.95 \pm 0.014$ | $0.96 \pm 0.008$ | $20 \pm 4$ |
| N: 35, P: 3, G: 70, F1 PR | $0.98 \pm 0.005$ | $0.94 \pm 0.011$ | $0.96 \pm 0.006$ | $7 \pm 0$ |
| N: 20, P: 3, G: 40, F1 PR | $0.98 \pm 0.005$ | $0.94 \pm 0.010$ | $0.96 \pm 0.006$ | $7 \pm 0$ |
| N: 10, P: 3, G: 20, F1 PR | $0.99 \pm 0.002$ | $0.93 \pm 0.011$ | $0.96 \pm 0.006$ | $7 \pm 0$ |
| N: 25, P: 4, G: 75, Coverage | $0.95 \pm 0.005$ | $0.96 \pm 0.012$ | $0.96 \pm 0.007$ | $7 \pm 0$ |

Table 6: The performance of each evaluation metric across all experiments grouped by the dataset used. Part A.

(a) Ego dataset.

| METRIC | FIDELITY | DIVERSITY | FIDELITY & DIVERSITY (RANDOM/PRETRAINED) | | SAMPLE EFF. (RANDOM/PRETRAINED) | |
|---|---|---|---|---|---|---|
| Orbits MMD | $0.200 \pm 0.053$ | $0.250 \pm 0.029$ | $0.220 \pm 0.030$ | | $319.000 \pm 21.000$ | |
| Degree MMD | $1.000 \pm 0.000$ | $0.760 \pm 0.083$ | $0.880 \pm 0.045$ | | $7.000 \pm 0.000$ | |
| Clustering MMD | $1.000 \pm 0.000$ | $0.550 \pm 0.120$ | $0.770 \pm 0.069$ | | $7.000 \pm 0.000$ | |
| NPSDK MMD | $1.000 \pm 0.000$ | $1.000 \pm 0.001$ | $1.000 \pm 0.001$ | | $7.000 \pm 0.000$ | |
| FD | $0.990 \pm 0.001$ | $0.490 \pm 0.028$ | $0.740 \pm 0.016$ | $0.910 \pm 0.009$ | $11.000 \pm 1.000$ | $12.000 \pm 2.000$ |
| KD | $0.990 \pm 0.001$ | $0.310 \pm 0.033$ | $0.650 \pm 0.020$ | $0.820 \pm 0.013$ | $18.000 \pm 1.000$ | $14.000 \pm 2.000$ |
| Precision | $0.880 \pm 0.009$ | $-0.480 \pm 0.029$ | $0.200 \pm 0.029$ | $0.250 \pm 0.030$ | $7.000 \pm 0.000$ | $7.000 \pm 0.000$ |
| Recall | $0.860 \pm 0.009$ | $0.970 \pm 0.004$ | $0.910 \pm 0.005$ | $0.960 \pm 0.002$ | $7.000 \pm 0.000$ | $7.000 \pm 0.000$ |
| Density | $0.830 \pm 0.011$ | $-0.180 \pm 0.031$ | $0.320 \pm 0.024$ | $0.360 \pm 0.027$ | $7.000 \pm 0.000$ | $7.000 \pm 0.000$ |
| Coverage | $0.910 \pm 0.005$ | $0.990 \pm 0.004$ | $0.950 \pm 0.003$ | $0.990 \pm 0.001$ | $7.000 \pm 0.000$ | $7.000 \pm 0.000$ |
| F1 PR | $0.900 \pm 0.007$ | $0.970 \pm 0.004$ | $0.930 \pm 0.004$ | $0.980 \pm 0.001$ | $7.000 \pm 0.000$ | $7.000 \pm 0.000$ |
| F1 DC | $0.880 \pm 0.007$ | $0.950 \pm 0.005$ | $0.920 \pm 0.004$ | $0.970 \pm 0.003$ | $7.000 \pm 0.000$ | $7.000 \pm 0.000$ |
| MMD Linear | $0.990 \pm 0.002$ | $0.420 \pm 0.028$ | $0.700 \pm 0.017$ | $0.880 \pm 0.010$ | $18.000 \pm 2.000$ | $8.000 \pm 1.000$ |
| MMD RBF | $0.960 \pm 0.006$ | $0.970 \pm 0.004$ | $0.970 \pm 0.004$ | $0.980 \pm 0.002$ | $14.000 \pm 1.000$ | $7.000 \pm 0.000$ |
| Average | $0.920 \pm 0.019$ | $0.540 \pm 0.168$ | $0.730 \pm 0.086$ | $0.810 \pm 0.086$ | $10.300 \pm 1.484$ | $8.300 \pm 0.803$ |

(b) Grid dataset.

| METRIC | FIDELITY | DIVERSITY | FIDELITY & DIVERSITY (RANDOM/PRETRAINED) | | SAMPLE EFF. (RANDOM/PRETRAINED) | |
|---|---|---|---|---|---|---|
| Orbits MMD | $0.290 \pm 0.156$ | $0.440 \pm 0.153$ | $0.350 \pm 0.109$ | | $9.000 \pm 2.000$ | |
| Degree MMD | $1.000 \pm 0.000$ | $0.290 \pm 0.177$ | $0.680 \pm 0.098$ | | $7.000 \pm 0.000$ | |
| Clustering MMD | $0.990 \pm 0.004$ | $0.000 \pm 0.000$ | $0.550 \pm 0.083$ | | $7.000 \pm 0.000$ | |
| NPSDK MMD | $0.990 \pm 0.004$ | $0.270 \pm 0.136$ | $0.630 \pm 0.088$ | | $7.000 \pm 0.000$ | |
| FD | $1.000 \pm 0.000$ | $0.070 \pm 0.034$ | $0.540 \pm 0.023$ | $0.550 \pm 0.023$ | $9.000 \pm 0.000$ | $7.000 \pm 0.000$ |
| KD | $0.970 \pm 0.007$ | $0.230 \pm 0.033$ | $0.600 \pm 0.021$ | $0.620 \pm 0.019$ | $9.000 \pm 0.000$ | $7.000 \pm 0.000$ |
| Precision | $0.960 \pm 0.003$ | $0.000 \pm 0.000$ | $0.480 \pm 0.017$ | $0.370 \pm 0.015$ | $7.000 \pm 0.000$ | $7.000 \pm 0.000$ |
| Recall | $0.500 \pm 0.015$ | $0.940 \pm 0.004$ | $0.720 \pm 0.011$ | $0.660 \pm 0.012$ | $7.000 \pm 0.000$ | $7.000 \pm 0.000$ |
| Density | $0.960 \pm 0.004$ | $0.180 \pm 0.033$ | $0.570 \pm 0.022$ | $0.450 \pm 0.021$ | $7.000 \pm 0.000$ | $7.000 \pm 0.000$ |
| Coverage | $0.840 \pm 0.012$ | $0.930 \pm 0.006$ | $0.890 \pm 0.007$ | $0.810 \pm 0.008$ | $7.000 \pm 0.000$ | $7.000 \pm 0.000$ |
| F1 PR | $0.960 \pm 0.003$ | $0.940 \pm 0.004$ | $0.950 \pm 0.003$ | $0.840 \pm 0.008$ | $7.000 \pm 0.000$ | $7.000 \pm 0.000$ |
| F1 DC | $0.950 \pm 0.005$ | $0.820 \pm 0.014$ | $0.890 \pm 0.008$ | $0.780 \pm 0.010$ | $7.000 \pm 0.000$ | $7.000 \pm 0.000$ |
| MMD Linear | $1.000 \pm 0.000$ | $0.030 \pm 0.029$ | $0.510 \pm 0.023$ | $0.520 \pm 0.022$ | $9.000 \pm 0.000$ | $7.000 \pm 0.000$ |
| MMD RBF | $0.950 \pm 0.007$ | $0.970 \pm 0.004$ | $0.960 \pm 0.004$ | $0.960 \pm 0.004$ | $9.000 \pm 1.000$ | $7.000 \pm 0.000$ |
| Average | $0.910 \pm 0.048$ | $0.510 \pm 0.138$ | $0.710 \pm 0.061$ | $0.660 \pm 0.060$ | $7.800 \pm 0.327$ | $7.000 \pm 0.000$ |

(c) Lobster dataset.

| METRIC | FIDELITY | DIVERSITY | FIDELITY & DIVERSITY (RANDOM/PRETRAINED) | | SAMPLE EFF. (RANDOM/PRETRAINED) | |
|---|---|---|---|---|---|---|
| Orbits MMD | $0.450 \pm 0.037$ | $0.150 \pm 0.079$ | $0.300 \pm 0.049$ | | $22.000 \pm 5.000$ | |
| Degree MMD | $1.000 \pm 0.001$ | $0.270 \pm 0.138$ | $0.630 \pm 0.090$ | | $7.000 \pm 0.000$ | |
| Clustering MMD | $0.950 \pm 0.013$ | $0.000 \pm 0.000$ | $0.480 \pm 0.076$ | | $7.000 \pm 0.000$ | |
| NPSDK MMD | $1.000 \pm 0.000$ | $0.920 \pm 0.028$ | $0.960 \pm 0.015$ | | $7.000 \pm 0.000$ | |
| FD | $0.990 \pm 0.001$ | $0.360 \pm 0.026$ | $0.680 \pm 0.017$ | $0.740 \pm 0.015$ | $11.000 \pm 1.000$ | $7.000 \pm 0.000$ |
| KD | $0.940 \pm 0.007$ | $0.240 \pm 0.023$ | $0.590 \pm 0.017$ | $0.620 \pm 0.018$ | $12.000 \pm 1.000$ | $9.000 \pm 1.000$ |
| Precision | $0.970 \pm 0.003$ | $-0.010 \pm 0.006$ | $0.480 \pm 0.018$ | $0.460 \pm 0.021$ | $7.000 \pm 0.000$ | $7.000 \pm 0.000$ |
| Recall | $0.570 \pm 0.016$ | $0.820 \pm 0.011$ | $0.700 \pm 0.011$ | $0.590 \pm 0.015$ | $7.000 \pm 0.000$ | $7.000 \pm 0.000$ |
| Density | $0.980 \pm 0.002$ | $-0.180 \pm 0.031$ | $0.400 \pm 0.026$ | $0.400 \pm 0.025$ | $7.000 \pm 0.000$ | $7.000 \pm 0.000$ |
| Coverage | $0.900 \pm 0.006$ | $0.860 \pm 0.013$ | $0.880 \pm 0.007$ | $0.880 \pm 0.007$ | $7.000 \pm 0.000$ | $7.000 \pm 0.000$ |
| F1 PR | $0.970 \pm 0.003$ | $0.820 \pm 0.011$ | $0.900 \pm 0.006$ | $0.890 \pm 0.007$ | $7.000 \pm 0.000$ | $7.000 \pm 0.000$ |
| F1 DC | $0.980 \pm 0.002$ | $0.610 \pm 0.028$ | $0.790 \pm 0.015$ | $0.740 \pm 0.016$ | $7.000 \pm 0.000$ | $7.000 \pm 0.000$ |
| MMD Linear | $0.980 \pm 0.002$ | $0.260 \pm 0.025$ | $0.620 \pm 0.018$ | $0.690 \pm 0.016$ | $14.000 \pm 1.000$ | $8.000 \pm 0.000$ |
| MMD RBF | $0.980 \pm 0.002$ | $0.870 \pm 0.014$ | $0.930 \pm 0.007$ | $0.930 \pm 0.006$ | $12.000 \pm 1.000$ | $8.000 \pm 0.000$ |
| Average | $0.930 \pm 0.040$ | $0.460 \pm 0.122$ | $0.700 \pm 0.057$ | $0.690 \pm 0.057$ | $9.100 \pm 0.888$ | $7.400 \pm 0.221$ |

Table 7: The performance of each evaluation metric across all experiments grouped by the dataset used. Part B.

(a) Community dataset.

| METRIC | FIDELITY | DIVERSITY | FIDELITY & DIVERSITY (RANDOM/PRETRAINED) | | SAMPLE EFF. (RANDOM/PRETRAINED) | |
|---|---|---|---|---|---|---|
| Orbits MMD | $0.760 \pm 0.065$ | $1.000 \pm 0.000$ | $0.880 \pm 0.038$ | | $79.000 \pm 31.000$ | |
| Degree MMD | $1.000 \pm 0.000$ | $0.960 \pm 0.011$ | $0.980 \pm 0.006$ | | $18.000 \pm 7.000$ | |
| Clustering MMD | $1.000 \pm 0.000$ | $0.810 \pm 0.061$ | $0.900 \pm 0.034$ | | $7.000 \pm 0.000$ | |
| NSPDK MMD | $0.997 \pm 0.001$ | $0.984 \pm 0.009$ | $0.991 \pm 0.004$ | | $11.545 \pm 3.049$ | |
| FD | $0.930 \pm 0.008$ | $0.440 \pm 0.029$ | $0.680 \pm 0.018$ | $0.600 \pm 0.018$ | $115.000 \pm 6.000$ | $188.000 \pm 6.000$ |
| KD | $-0.380 \pm 0.030$ | $0.300 \pm 0.033$ | $-0.040 \pm 0.025$ | $0.130 \pm 0.021$ | $180.000 \pm 6.000$ | $202.000 \pm 5.000$ |
| Precision | $0.340 \pm 0.019$ | $-0.350 \pm 0.022$ | $-0.000 \pm 0.019$ | $0.120 \pm 0.028$ | $7.000 \pm 0.000$ | $7.000 \pm 0.000$ |
| Recall | $0.790 \pm 0.018$ | $0.980 \pm 0.004$ | $0.890 \pm 0.010$ | $0.960 \pm 0.003$ | $7.000 \pm 0.000$ | $7.000 \pm 0.000$ |
| Density | $0.770 \pm 0.013$ | $-0.350 \pm 0.033$ | $0.210 \pm 0.027$ | $0.250 \pm 0.027$ | $7.000 \pm 0.000$ | $7.000 \pm 0.000$ |
| Coverage | $0.920 \pm 0.007$ | $0.990 \pm 0.004$ | $0.950 \pm 0.004$ | $0.980 \pm 0.002$ | $7.000 \pm 0.000$ | $7.000 \pm 0.000$ |
| F1 PR | $0.800 \pm 0.018$ | $0.980 \pm 0.004$ | $0.890 \pm 0.009$ | $0.980 \pm 0.002$ | $7.000 \pm 0.000$ | $7.000 \pm 0.000$ |
| F1 DC | $0.960 \pm 0.006$ | $0.950 \pm 0.005$ | $0.950 \pm 0.004$ | $0.960 \pm 0.004$ | $7.000 \pm 0.000$ | $7.000 \pm 0.000$ |
| MMD Linear | $0.940 \pm 0.007$ | $0.410 \pm 0.023$ | $0.670 \pm 0.015$ | $0.710 \pm 0.016$ | $102.000 \pm 6.000$ | $110.000 \pm 6.000$ |
| MMD RBF | $0.990 \pm 0.004$ | $0.970 \pm 0.004$ | $0.980 \pm 0.003$ | $0.980 \pm 0.003$ | $91.000 \pm 6.000$ | $30.000 \pm 3.000$ |
| Average | $0.710 \pm 0.135$ | $0.530 \pm 0.170$ | $0.620 \pm 0.129$ | $0.670 \pm 0.117$ | $53.000 \pm 20.142$ | $57.200 \pm 25.105$ |

(b) Proteins dataset.

| METRIC | FIDELITY | DIVERSITY | FIDELITY & DIVERSITY (RANDOM/PRETRAINED) | | SAMPLE EFF. (RANDOM/PRETRAINED) | |
|---|---|---|---|---|---|---|
| Orbits MMD | $0.220 \pm 0.126$ | $0.630 \pm 0.072$ | $0.420 \pm 0.079$ | | $177.000 \pm 67.000$ | |
| Degree MMD | $1.000 \pm 0.000$ | $0.290 \pm 0.139$ | $0.640 \pm 0.089$ | | $7.000 \pm 0.000$ | |
| Clustering MMD | $0.990 \pm 0.001$ | $0.760 \pm 0.046$ | $0.880 \pm 0.029$ | | $7.000 \pm 0.000$ | |
| NSPDK MMD | $1.000 \pm 0.000$ | $0.998 \pm 0.001$ | $0.999 \pm 0.001$ | | $7.000 \pm 0.000$ | |
| FD | $0.990 \pm 0.003$ | $0.840 \pm 0.011$ | $0.920 \pm 0.006$ | $0.900 \pm 0.010$ | $150.000 \pm 13.000$ | $60.000 \pm 9.000$ |
| KD | $0.550 \pm 0.033$ | $0.550 \pm 0.025$ | $0.550 \pm 0.021$ | $0.710 \pm 0.018$ | $236.000 \pm 15.000$ | $83.000 \pm 11.000$ |
| Precision | $0.970 \pm 0.006$ | $-0.430 \pm 0.029$ | $0.270 \pm 0.029$ | $0.250 \pm 0.029$ | $7.000 \pm 0.000$ | $7.000 \pm 0.000$ |
| Recall | $0.790 \pm 0.013$ | $0.950 \pm 0.005$ | $0.870 \pm 0.008$ | $0.890 \pm 0.005$ | $7.000 \pm 0.000$ | $7.000 \pm 0.000$ |
| Density | $0.990 \pm 0.004$ | $0.020 \pm 0.031$ | $0.500 \pm 0.024$ | $0.370 \pm 0.027$ | $7.000 \pm 0.000$ | $7.000 \pm 0.000$ |
| Coverage | $0.970 \pm 0.004$ | $0.990 \pm 0.004$ | $0.980 \pm 0.003$ | $0.990 \pm 0.000$ | $7.000 \pm 0.000$ | $7.000 \pm 0.000$ |
| F1 PR | $0.980 \pm 0.005$ | $0.930 \pm 0.006$ | $0.950 \pm 0.004$ | $0.960 \pm 0.003$ | $7.000 \pm 0.000$ | $7.000 \pm 0.000$ |
| F1 DC | $0.990 \pm 0.002$ | $0.960 \pm 0.005$ | $0.970 \pm 0.003$ | $0.970 \pm 0.002$ | $7.000 \pm 0.000$ | $7.000 \pm 0.000$ |
| MMD Linear | $0.990 \pm 0.002$ | $0.780 \pm 0.017$ | $0.880 \pm 0.009$ | $0.940 \pm 0.006$ | $145.000 \pm 12.000$ | $21.000 \pm 4.000$ |
| MMD RBF | $0.990 \pm 0.002$ | $0.960 \pm 0.004$ | $0.970 \pm 0.002$ | $0.990 \pm 0.001$ | $86.000 \pm 8.000$ | $8.000 \pm 1.000$ |
| Average | $0.920 \pm 0.046$ | $0.660 \pm 0.153$ | $0.790 \pm 0.080$ | $0.800 \pm 0.086$ | $65.900 \pm 26.559$ | $21.400 \pm 8.634$ |

(c) Zinc dataset. Note that the metrics from You et al. (2018) are excluded here as they cannot incorporate node and edge features in evaluation.

| METRIC | FIDELITY | DIVERSITY | FIDELITY & DIVERSITY | NODE/EDGE FEATS. | SAMPLE EFF. |
|---|---|---|---|---|---|
| NPSDK MMD | $1.000 \pm 0.000$ | $1.000 \pm 0.000$ | $1.000 \pm 0.000$ | $1.000 \pm 0.000$ | $7 \pm 0$ |
| FD | $0.950 \pm 0.006$ | $0.690 \pm 0.031$ | $0.820 \pm 0.017$ | $0.960 \pm 0.010$ | $35.000 \pm 7.000$ |
| KD | $0.910 \pm 0.010$ | $0.540 \pm 0.035$ | $0.730 \pm 0.020$ | $0.940 \pm 0.011$ | $39.000 \pm 7.000$ |
| Precision | $0.980 \pm 0.005$ | $-0.410 \pm 0.035$ | $0.290 \pm 0.034$ | $0.990 \pm 0.001$ | $7.000 \pm 0.000$ |
| Recall | $0.940 \pm 0.005$ | $0.980 \pm 0.002$ | $0.960 \pm 0.003$ | $0.800 \pm 0.018$ | $7.000 \pm 0.000$ |
| Density | $1.000 \pm 0.001$ | $-0.450 \pm 0.035$ | $0.270 \pm 0.035$ | $0.990 \pm 0.001$ | $7.000 \pm 0.000$ |
| Coverage | $0.990 \pm 0.000$ | $1.000 \pm 0.000$ | $0.990 \pm 0.000$ | $0.990 \pm 0.000$ | $7.000 \pm 0.000$ |
| F1 PR | $0.990 \pm 0.002$ | $0.930 \pm 0.006$ | $0.960 \pm 0.003$ | $0.990 \pm 0.000$ | $7.000 \pm 0.000$ |
| F1 DC | $1.000 \pm 0.000$ | $0.850 \pm 0.011$ | $0.920 \pm 0.006$ | $0.990 \pm 0.000$ | $7.000 \pm 0.000$ |
| MMD Linear | $0.990 \pm 0.002$ | $0.810 \pm 0.018$ | $0.900 \pm 0.010$ | $0.990 \pm 0.005$ | $16.000 \pm 3.000$ |
| MMD RBF | $1.000 \pm 0.001$ | $1.000 \pm 0.000$ | $1.000 \pm 0.000$ | $1.000 \pm 0.001$ | $9.000 \pm 1.000$ |
| Average | $0.980 \pm 0.010$ | $0.590 \pm 0.177$ | $0.780 \pm 0.088$ | $0.960 \pm 0.019$ | $14.100 \pm 3.928$ |

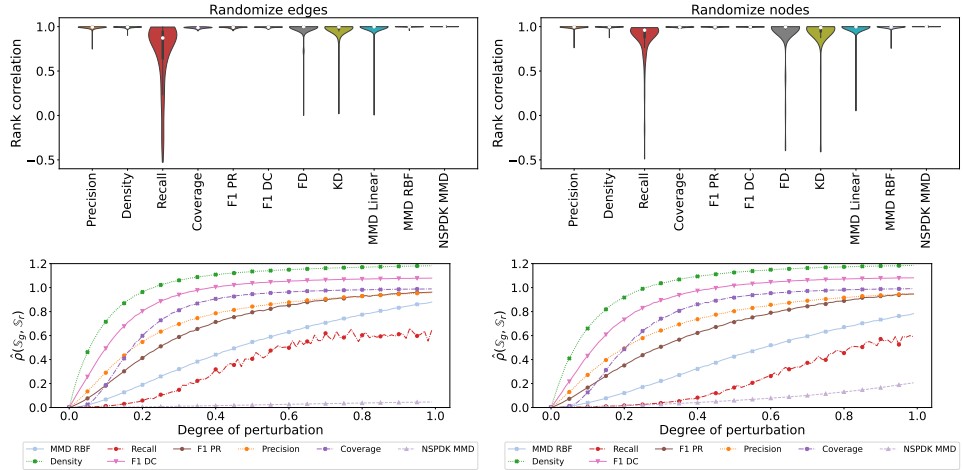

Figure 9: Results from the randomizing edge features (left) and randomizing node features (right) experiments on the ZINC dataset.

Table 8: Comparing the performance of metrics on the mixing experiment when the mixed graphs are random E-R graphs or graphs generated by GRAN (Liao et al., 2019).

| METRIC | MIXING RANDOM | MIXING GENERATED |
|---|---|---|
| Orbits MMD | $0.680 \pm 0.047$ | $0.010 \pm 0.035$ |
| Degree MMD | $1.000 \pm 0.000$ | $1.000 \pm 0.001$ |
| Clustering MMD | $1.000 \pm 0.000$ | $0.980 \pm 0.005$ |
| NPSDK MMD | $1.000 \pm 0.000$ | $1.000 \pm 0.000$ |
| FD | $0.970 \pm 0.003$ | $0.980 \pm 0.003$ |
| KD | $0.730 \pm 0.016$ | $0.780 \pm 0.018$ |
| Precision | $0.920 \pm 0.007$ | $0.970 \pm 0.005$ |
| Recall | $0.640 \pm 0.011$ | $0.490 \pm 0.012$ |
| Density | $1.000 \pm 0.001$ | $0.990 \pm 0.002$ |
| Coverage | $0.920 \pm 0.004$ | $0.900 \pm 0.005$ |
| F1 PR | $0.960 \pm 0.006$ | $0.970 \pm 0.005$ |
| F1 DC | $1.000 \pm 0.001$ | $0.990 \pm 0.002$ |
| MMD Linear | $0.970 \pm 0.003$ | $0.960 \pm 0.004$ |
| MMD RBF | $1.000 \pm 0.001$ | $0.990 \pm 0.002$ |
| Average | $0.910 \pm 0.039$ | $0.900 \pm 0.050$ |

## C.4 MIXING GENERATED GRAPHS

As opposed to the E-R graphs used in Section 4.1, here we perform the mixing experiment using graphs generated by GRAN (Liao et al., 2019) for each dataset. As these graphs are a better representation of the dataset of interest, this represents a much harder problem. However, as seen in Table 8, the results across metrics are fairly consistent regardless of the type of graph used.

## C.5 COMPUTATIONAL EFFICIENCY

In Figure 12 we plot each metric's RAM usage throughout the computational efficiency experiments in Section 4.5 by tracking the memory usage of the main Python process. While this is likely imperfect and slightly noisy, it provides an idea of the memory requirements of each metric. Unfortunately all classifier-based metrics are grouped into the same experiment and more in-depth breakdowns are unavailable for this experiment. However, it is likely that the bulk of the memory requirements spe-

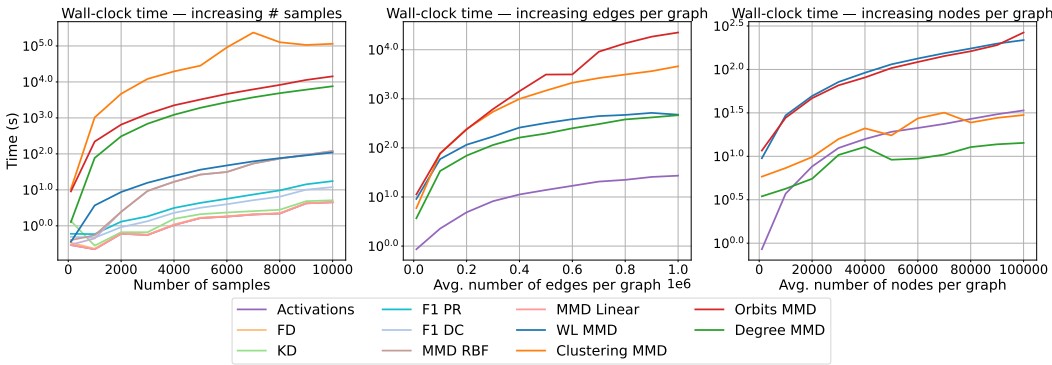

Figure 10: The wall-clock time of each metric as datasets scale in a single dimension. Activations is the time to extract graph embeddings from GIN on a **CPU**.

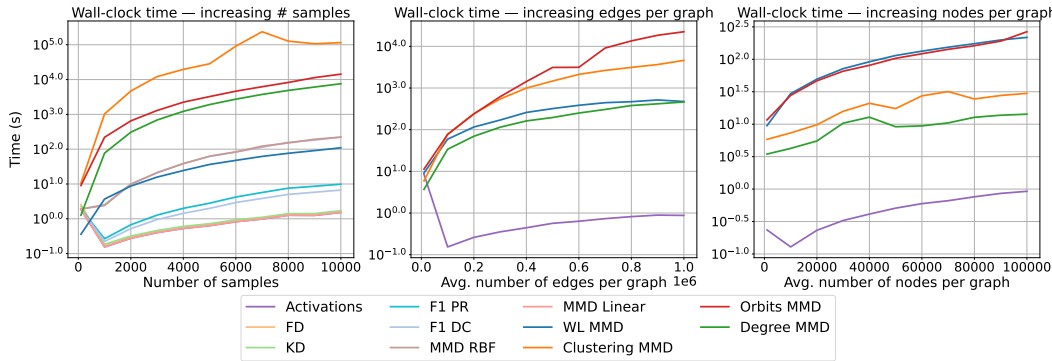

Figure 11: The wall-clock time of each metric as datasets scale in a single dimension. Activations is the time to extract graph embeddings from GIN on a **GPU**.

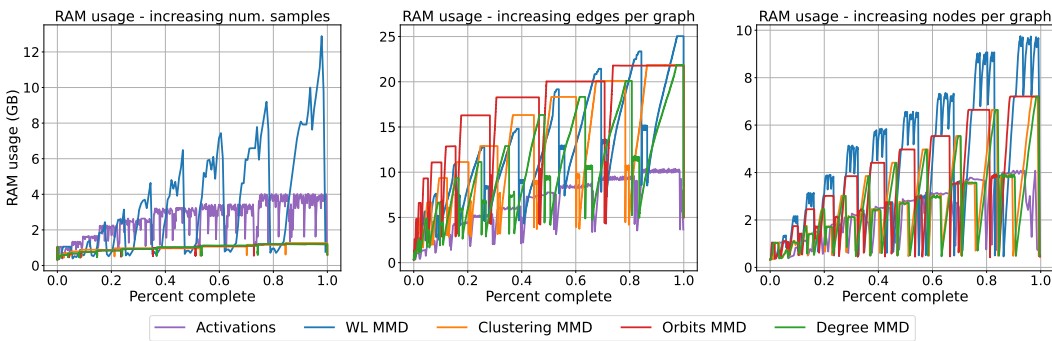

Figure 12: The estimated memory usage of each metric as the dataset scales in a single dimension. Activations is the memory required for all classifier-based metrics combined.

cific to these metrics come from storing the GIN parameters and extracted graph embeddings. The maximum recorded value for each metric in each experiment is recorded in Table 9.

## C.6 COMPARING OTHER GNNS

In the main article, all experiments were performed using GIN with summation neighborhood aggregation and concatenating graph embeddings at each round of graph propagation into a final graph embedding (Equation 4). Here we also provide results for random GINs with mean and max

Table 9: The maximum recorded memory usage in GB of each metric during each of the computational efficiency experiments.

| METRIC | SCALING EXPERIMENT | | |
|---|---|---|---|
| | Num. samples | Nodes | Edges |
| Degree MMD | 1.20 | 7.20 | 21.88 |
| Clustering MMD | 1.26 | 7.20 | 21.85 |
| Orbits MMD | 1.19 | 7.20 | 21.80 |
| Classifier metrics | 4.01 | 4.08 | 10.40 |

Table 10: Comparing various GNN architectures on our experiments using all datasets. We present only the final two recommended metrics to keep this presentation concise. All models are randomly initialized and aggregated across the same underlying model architectures (number of propagation rounds, node embedding size) used in the main article.

| METRIC | FIDELITY | DIVERSITY | FIDELITY & DIVERSITY | NODE/EDGE FEATS. | SAMPLE EFF. |
|---|---|---|---|---|---|
| F1 PR (GraphSAGE) | $0.900 \pm 0.004$ | $0.920 \pm 0.004$ | $0.910 \pm 0.003$ | $0.990 \pm 0.001$ | $7 \pm 0$ |
| F1 PR (GCN) | $0.870 \pm 0.005$ | $0.910 \pm 0.005$ | $0.890 \pm 0.003$ | $0.990 \pm 0.000$ | $7 \pm 0$ |
| F1 PR (GIN, no concat.) | $0.790 \pm 0.007$ | $0.920 \pm 0.004$ | $0.860 \pm 0.004$ | $0.980 \pm 0.004$ | $7 \pm 0$ |
| F1 PR (GIN) | $0.920 \pm 0.004$ | $0.930 \pm 0.003$ | $0.930 \pm 0.003$ | $0.990 \pm 0.000$ | $7 \pm 0$ |
| MMD RBF (GraphSAGE) | $0.950 \pm 0.003$ | $0.950 \pm 0.003$ | $0.950 \pm 0.002$ | $1.000 \pm 0.001$ | $23 \pm 2$ |
| MMD RBF (GCN) | $0.950 \pm 0.003$ | $0.950 \pm 0.003$ | $0.950 \pm 0.002$ | $1.000 \pm 0.002$ | $55 \pm 3$ |
| MMD RBF (GIN, no concat.) | $0.960 \pm 0.003$ | $0.940 \pm 0.003$ | $0.950 \pm 0.002$ | $0.990 \pm 0.005$ | $60 \pm 3$ |
| MMD RBF (GIN) | $0.970 \pm 0.002$ | $0.950 \pm 0.003$ | $0.960 \pm 0.002$ | $1.000 \pm 0.001$ | $42 \pm 2$ |

neighborhood aggregation. With minor architecture differences, the former is equivalent to random GCN (Kipf & Welling, 2017), while the latter is equivalent to random GraphSAGE (Hamilton et al., 2018). In addition, we also provide a comparison to random GIN with summation neighborhood aggregation that does not use the concatenation in Equation 4, i.e. the resultant graph embedding is simply the embedding obtained at graph propagation round $L$. We find that while GIN is superior across the presented metrics, it is in many cases only by a very narrow margin (Table 10). The performance of MMD RBF appears to be relatively constant across GNNs, while the performance of F1 PR has higher variance.

## C.7 COMPARING $\sigma$ SELECTION STRATEGIES

In order to provide a fairer comparison to pre-existing metrics based on the RBF kernel (You et al., 2018), here we include results for MMD RBF with a static $\sigma = 1$ using a random GIN. The results are shown in Table 11. While this metric does result in a decrease in performance while measuring diversity, it is still more expressive than baseline metrics. Note that we could also utilize our $\sigma$ selection process with pre-existing metrics. However, we did not experiment with this as these metrics would still suffer from computational efficiency issues and be unable to incorporate node and edge features.

Table 11: Measuring the impact of our $\sigma$ selection process on the MMD RBF metric. While using a static $\sigma$ does result in a decrease in performance, it is still more expressive than pre-existing metrics that rely on the RBF kernel (You et al., 2018).

| METRIC | FIDELITY | DIVERSITY | FIDELITY & DIVERSITY | NODE/EDGE FEATS. | SAMPLE EFF. |
|---|---|---|---|---|---|
| Orbits MMD | $0.370 \pm 0.048$ | $0.490 \pm 0.046$ | $0.430 \pm 0.034$ | N/A | $122 \pm 22$ |
| Degree MMD | $1.000 \pm 0.000$ | $0.510 \pm 0.061$ | $0.760 \pm 0.035$ | N/A | $9 \pm 1$ |
| Clustering MMD | $0.990 \pm 0.003$ | $0.430 \pm 0.047$ | $0.720 \pm 0.030$ | N/A | $7 \pm 0$ |
| MMD RBF ($\sigma = 1$) | $0.970 \pm 0.002$ | $0.850 \pm 0.007$ | $0.910 \pm 0.004$ | $0.890 \pm 0.009$ | $33 \pm 3$ |
| MMD RBF | $0.970 \pm 0.002$ | $0.950 \pm 0.003$ | $0.960 \pm 0.002$ | $1.000 \pm 0.001$ | $42 \pm 2$ |

## C.8    GGM SELECTION

Here we evaluate GGMs at various stages of training on Grid, Lobster and Proteins datasets using classical metrics (You et al., 2018), NN-based metrics using a random GIN, and NN-based metrics using a pretrained GIN. The results are shown in Table 12.

Table 12: Evaluation of different GGMs at various percentages of total epochs trained on the Grid, Lobster, and Proteins datasets. Models are evaluated using three of the strongest NN-based metrics and classical metrics (You et al., 2018). All NN-based metrics are averaged across 10 different GINs with the strongest configuration. Cells are colored according to their rank in a given column. 50/50 split represents the metric computed using a random 50/50 split of the dataset and represents the theoretical ideal score for each metric.

(a) Random GIN, Grid dataset.

| | MMD RBF | F1 PR | F1 DC | Clus. | Deg. | Orbit |
|---|---|---|---|---|---|---|
| 50/50 split | 0.042 | 0.998 | 0.995 | 0.0 | $6.51e^{-5}$ | 0.018 |
| Erdős-Rényi | $0.203 \pm 0.006$ | $0.669 \pm 0.041$ | $0.89 \pm 0.026$ | 0.023 | 1.001 | 0.56 |
| GraphRNN-100% | $0.184 \pm 5e^{-5}$ | $0.919 \pm 1e-16$ | $0.896 \pm 0.002$ | $4.59e^{-8}$ | 0.032 | 0.252 |
| GraphRNN-66% | $0.154 \pm 7e^{-5}$ | $0.912 \pm 0.007$ | $0.942 \pm 7e^{-4}$ | $1.69e^{-6}$ | 0.015 | 0.169 |
| GraphRNN-33% | $0.248 \pm 6e^{-5}$ | $0.953 \pm 1e-16$ | $0.878 \pm 0.003$ | $1.81e^{-6}$ | 0.022 | 0.134 |
| GRAN-100% | $0.063 \pm 0.001$ | $1.0 \pm 0.0$ | $1.073 \pm 0.001$ | $1.24e^{-6}$ | $1.68e^{-4}$ | 0.037 |
| GRAN-66% | $0.061 \pm 0.001$ | $1.0 \pm 0.0$ | $1.065 \pm 0.002$ | $3.15e^{-6}$ | $3.20e^{-5}$ | 0.047 |
| GRAN-33% | $0.06 \pm 2e^{-4}$ | $0.982 \pm 0.012$ | $1.067 \pm 0.005$ | $1.46e^{-6}$ | $4.85e^{-5}$ | 0.054 |

(b) Pretrained GIN, Grid dataset.

| | MMD RBF | F1 PR | F1 DC | Clus. | Deg. | Orbit |
|---|---|---|---|---|---|---|
| 50/50 split | 0.05 | 0.997 | 0.969 | 0.0 | $6.51e^{-5}$ | 0.018 |
| Erdős-Rényi | $1.278 \pm 0.05$ | $0.0 \pm 0.0$ | $0.0 \pm 0.0$ | 0.023 | 1.001 | 0.56 |
| GraphRNN-100% | $0.21 \pm 0.017$ | $0.928 \pm 0.021$ | $0.82 \pm 0.03$ | $4.59e^{-8}$ | 0.032 | 0.252 |
| GraphRNN-66% | $0.156 \pm 0.012$ | $0.907 \pm 0.018$ | $0.871 \pm 0.033$ | $1.69e^{-6}$ | 0.015 | 0.169 |
| GraphRNN-33% | $0.176 \pm 0.009$ | $0.951 \pm 0.015$ | $0.848 \pm 0.035$ | $1.81e^{-6}$ | 0.022 | 0.134 |
| GRAN-100% | $0.078 \pm 0.016$ | $0.98 \pm 0.017$ | $1.004 \pm 0.05$ | $1.24e^{-6}$ | $1.68e^{-4}$ | 0.037 |
| GRAN-66% | $0.066 \pm 0.009$ | $0.963 \pm 0.032$ | $0.994 \pm 0.052$ | $3.15e^{-6}$ | $3.20e^{-5}$ | 0.047 |
| GRAN-33% | $0.073 \pm 0.015$ | $0.953 \pm 0.029$ | $0.99 \pm 0.052$ | $1.46e^{-6}$ | $4.85e^{-5}$ | 0.054 |

(c) Random GIN, Lobster dataset

| | MMD RBF | F1 PR | F1 DC | Clus. | Deg. | Orbit |
|---|---|---|---|---|---|---|
| 50/50 split | 0.049 | 0.989 | 0.987 | 0.0 | 0.003 | 0.037 |
| Erdős-Rényi | $0.538 \pm 0.029$ | $0.446 \pm 0.126$ | $0.554 \pm 0.135$ | 0.002 | 0.736 | 0.729 |
| GraphRNN-100% | $0.235 \pm 0.005$ | $0.979 \pm 0.011$ | $0.921 \pm 0.026$ | 0.095 | 0.038 | 0.104 |
| GraphRNN-66% | $0.244 \pm 0.006$ | $0.983 \pm 0.008$ | $0.914 \pm 0.023$ | 0.099 | 0.05 | 0.104 |
| GraphRNN-33% | $0.137 \pm 0.01$ | $0.939 \pm 0.056$ | $0.893 \pm 0.031$ | 0.137 | 0.047 | 0.104 |
| GRAN-100% | $0.146 \pm 0.009$ | $0.932 \pm 0.041$ | $0.905 \pm 0.022$ | 0.101 | 0.048 | 0.098 |
| GRAN-66% | $0.161 \pm 0.01$ | $0.956 \pm 0.007$ | $0.942 \pm 0.022$ | 0.103 | 0.045 | 0.094 |
| GRAN-33% | $0.096 \pm 0.013$ | $0.921 \pm 0.054$ | $0.907 \pm 0.032$ | 0.125 | 0.08 | 0.099 |

(d) Pretrained GIN, Lobster dataset

| | MMD RBF | F1 PR | F1 DC | Clus. | Deg. | Orbit |
|---|---|---|---|---|---|---|
| 50/50 split | 0.043 | 0.97 | 0.977 | 0.0 | 0.003 | 0.037 |
| Erdős-Rényi | $0.676 \pm 0.23$ | $0.088 \pm 0.085$ | $0.077 \pm 0.077$ | 0.002 | 0.736 | 0.729 |
| GraphRNN-100% | $0.193 \pm 0.026$ | $0.7 \pm 0.058$ | $0.658 \pm 0.037$ | 0.095 | 0.038 | 0.104 |
| GraphRNN-66% | $0.222 \pm 0.039$ | $0.671 \pm 0.06$ | $0.582 \pm 0.039$ | 0.099 | 0.05 | 0.104 |
| GraphRNN-33% | $0.196 \pm 0.033$ | $0.652 \pm 0.041$ | $0.615 \pm 0.043$ | 0.137 | 0.047 | 0.104 |
| GRAN-100% | $0.153 \pm 0.02$ | $0.698 \pm 0.04$ | $0.661 \pm 0.042$ | 0.101 | 0.048 | 0.098 |
| GRAN-66% | $0.186 \pm 0.025$ | $0.68 \pm 0.051$ | $0.602 \pm 0.032$ | 0.103 | 0.045 | 0.094 |
| GRAN-33% | $0.194 \pm 0.016$ | $0.595 \pm 0.035$ | $0.565 \pm 0.051$ | 0.125 | 0.08 | 0.099 |

(e) Random GIN, Proteins dataset.

| | MMD RBF | F1 PR | F1 DC | Clus. | Deg. | Orbit |
|---|---|---|---|---|---|---|
| 50/50 split | 0.005 | 0.988 | 0.988 | $7.18e^{-4}$ | $8.64e^{-4}$ | 0.004 |
| Erdős-Rényi | $0.733 \pm 0.099$ | $0.023 \pm 0.011$ | $0.01 \pm 0.004$ | 1.796 | 1.523 | 0.124 |
| GRAN-100% | $0.186 \pm 0.065$ | $0.756 \pm 0.1$ | $0.568 \pm 0.114$ | 0.28 | 0.542 | 0.099 |
| GRAN-75% | $0.09 \pm 0.034$ | $0.853 \pm 0.055$ | $0.819 \pm 0.1$ | 0.323 | 0.304 | 0.044 |
| GRAN-50% | $0.226 \pm 0.071$ | $0.66 \pm 0.109$ | $0.474 \pm 0.111$ | 0.382 | 0.646 | 0.103 |
| GRAN-25% | $0.019 \pm 0.009$ | $0.922 \pm 0.004$ | $0.995 \pm 0.045$ | 0.291 | 0.077 | 0.028 |

(f) Pretrained GIN, Proteins dataset

| | MMD RBF | F1 PR | F1 DC | Clus. | Deg. | Orbit |
|---|---|---|---|---|---|---|
| 50/50 split | 0.005 | 0.973 | 0.987 | $7.18e^{-4}$ | $8.64e^{-4}$ | 0.004 |
| Erdős-Rényi | $1.097 \pm 0.102$ | $4.34e^{-4} \pm 9e^{-4}$ | $1.68e^{-4} \pm 3e^{-4}$ | 1.796 | 1.523 | 0.124 |
| GRAN-100% | $0.514 \pm 0.083$ | $0.561 \pm 0.141$ | $0.306 \pm 0.085$ | 0.28 | 0.542 | 0.099 |
| GRAN-75% | $0.304 \pm 0.061$ | $0.691 \pm 0.074$ | $0.488 \pm 0.072$ | 0.323 | 0.304 | 0.044 |
| GRAN-50% | $0.549 \pm 0.086$ | $0.463 \pm 0.13$ | $0.228 \pm 0.076$ | 0.382 | 0.646 | 0.103 |
| GRAN-25% | $0.066 \pm 0.014$ | $0.885 \pm 0.022$ | $0.83 \pm 0.033$ | 0.291 | 0.077 | 0.028 |

Table 13: Evaluating GGMs at various stages of training on the **Proteins** dataset using all 20 GIN architectures tested in the main-body.

(a) Using the MMD RBF metric. First 10 archictectures.

|            | 0     | 1     | 2     | 3     | 4     | 5     | 6     | 7     | 8     | 9     |
|------------|-------|-------|-------|-------|-------|-------|-------|-------|-------|-------|
| 50/50 split| 0.004 | 0.005 | 0.005 | 0.005 | 0.005 | 0.005 | 0.005 | 0.004 | 0.005 | 0.005 |
| GRAN-100%  | 0.593 | 0.238 | 0.127 | 0.365 | 0.122 | 0.161 | 0.363 | 0.427 | 0.161 | 0.447 |
| GRAN-75%   | 0.334 | 0.126 | 0.066 | 0.073 | 0.063 | 0.071 | 0.203 | 0.245 | 0.19  | 0.195 |
| GRAN-50%   | 0.62  | 0.234 | 0.159 | 0.187 | 0.152 | 0.172 | 0.356 | 0.421 | 0.39  | 0.419 |
| GRAN-25%   | 0.068 | 0.037 | 0.013 | 0.034 | 0.012 | 0.026 | 0.042 | 0.054 | 0.042 | 0.044 |

(b) Using the MMD RBF metric. Part B

|            | 10    | 11    | 12    | 13    | 14    | 15    | 16    | 17    | 18    | 19    |
|------------|-------|-------|-------|-------|-------|-------|-------|-------|-------|-------|
| 50/50 split| 0.005 | 0.005 | 0.005 | 0.005 | 0.005 | 0.005 | 0.005 | 0.005 | 0.004 | 0.005 |
| GRAN-100%  | 0.186 | 0.307 | 0.474 | 0.362 | 0.134 | 0.307 | 0.211 | 0.431 | 0.358 | 0.277 |
| GRAN-75%   | 0.09  | 0.127 | 0.242 | 0.17  | 0.069 | 0.121 | 0.084 | 0.158 | 0.152 | 0.11  |
| GRAN-50%   | 0.226 | 0.329 | 0.493 | 0.361 | 0.168 | 0.319 | 0.234 | 0.18  | 0.36  | 0.302 |
| GRAN-25%   | 0.019 | 0.031 | 0.053 | 0.037 | 0.013 | 0.033 | 0.026 | 0.033 | 0.04  | 0.028 |

(c) Using the F1 PR metric. Part A.

|            | 0     | 1     | 2     | 3     | 4     | 5     | 6     | 7     | 8     | 9     |
|------------|-------|-------|-------|-------|-------|-------|-------|-------|-------|-------|
| 50/50 split| 0.976 | 0.986 | 0.993 | 0.979 | 0.988 | 0.986 | 0.977 | 0.988 | 0.978 | 0.977 |
| GRAN-100%  | 0.313 | 0.714 | 0.75  | 0.678 | 0.659 | 0.802 | 0.517 | 0.702 | 0.539 | 0.661 |
| GRAN-75%   | 0.564 | 0.814 | 0.841 | 0.74  | 0.774 | 0.862 | 0.699 | 0.714 | 0.691 | 0.749 |
| GRAN-50%   | 0.254 | 0.637 | 0.719 | 0.577 | 0.529 | 0.737 | 0.459 | 0.571 | 0.48  | 0.585 |
| GRAN-25%   | 0.848 | 0.911 | 0.925 | 0.891 | 0.898 | 0.879 | 0.868 | 0.908 | 0.853 | 0.919 |

(d) Using the F1 PR metric. Part B.

|            | 10    | 11    | 12    | 13    | 14    | 15    | 16    | 17    | 18    | 19    |
|------------|-------|-------|-------|-------|-------|-------|-------|-------|-------|-------|
| 50/50 split| 0.988 | 0.991 | 0.978 | 0.98  | 0.992 | 0.975 | 0.986 | 0.983 | 0.978 | 0.986 |
| GRAN-100%  | 0.756 | 0.859 | 0.545 | 0.695 | 0.762 | 0.793 | 0.846 | 0.407 | 0.671 | 0.802 |
| GRAN-75%   | 0.853 | 0.922 | 0.654 | 0.772 | 0.819 | 0.836 | 0.913 | 0.614 | 0.767 | 0.894 |
| GRAN-50%   | 0.66  | 0.796 | 0.43  | 0.575 | 0.662 | 0.696 | 0.77  | 0.326 | 0.591 | 0.698 |
| GRAN-25%   | 0.922 | 0.962 | 0.879 | 0.901 | 0.915 | 0.924 | 0.933 | 0.874 | 0.88  | 0.949 |

## C.9 STABILITY OF GGM RANKINGS

Here, we investigate the stability of GGM rankings across GIN architectures. For the Grid, Lobster, and Proteins datasets, we again evaluate GGMs at various stages of training. We evaluate models using a random GIN with each of the 20 architectures we tested in the main-body using both MMD RBF and F1 PR, and the results are shown in Tables 13 through 15. We find that the rankings are extremely consistent across architectures for the Grid and Proteins datasets. However, this does not appear to be the case for the Lobster dataset. One potential explanation is that for the Lobster dataset, there is added difficulty from both small extremely sample sizes and poor generative models.

Table 14: Evaluating GGMs at various stages of training on the **Lobster** dataset using all 20 GIN architectures tested in the main-body.

(a) Using the MMD RBF metric. Part A.

|  | 0 | 1 | 2 | 3 | 4 | 5 | 6 | 7 | 8 | 9 |
|---|---|---|---|---|---|---|---|---|---|---|
| 50/50 split | 0.046 | 0.045 | 0.042 | 0.041 | 0.047 | 0.043 | 0.041 | 0.052 | 0.041 | 0.044 |
| GraphRNN-100% | 0.219 | 0.188 | 0.232 | 0.12 | 0.252 | 0.251 | 0.241 | 0.275 | 0.222 | 0.232 |
| GraphRNN-66% | 0.18 | 0.201 | 0.253 | 0.235 | 0.225 | 0.265 | 0.236 | 0.266 | 0.241 | 0.249 |
| GRAN-100% | 0.224 | 0.158 | 0.194 | 0.123 | 0.147 | 0.195 | 0.203 | 0.252 | 0.268 | 0.191 |
| GRAN-66% | 0.181 | 0.187 | 0.212 | 0.174 | 0.173 | 0.219 | 0.193 | 0.256 | 0.202 | 0.184 |

(b) Using the MMD RBF metric. Part B

|  | 10 | 11 | 12 | 13 | 14 | 15 | 16 | 17 | 18 | 19 |
|---|---|---|---|---|---|---|---|---|---|---|
| 50/50 split | 0.049 | 0.041 | 0.043 | 0.044 | 0.042 | 0.043 | 0.041 | 0.047 | 0.052 | 0.05 |
| GraphRNN-100% | 0.235 | 0.241 | 0.221 | 0.245 | 0.254 | 0.122 | 0.22 | 0.224 | 0.216 | 0.245 |
| GraphRNN-66% | 0.244 | 0.308 | 0.191 | 0.244 | 0.247 | 0.209 | 0.197 | 0.254 | 0.26 | 0.236 |
| GRAN-100% | 0.146 | 0.218 | 0.239 | 0.231 | 0.19 | 0.131 | 0.181 | 0.171 | 0.183 | 0.165 |
| GRAN-66% | 0.161 | 0.246 | 0.203 | 0.215 | 0.206 | 0.174 | 0.189 | 0.192 | 0.207 | 0.186 |

(c) Using the F1 PR metric. Part A.

|  | 0 | 1 | 2 | 3 | 4 | 5 | 6 | 7 | 8 | 9 |
|---|---|---|---|---|---|---|---|---|---|---|
| 50/50 split | 0.983 | 0.997 | 0.996 | 0.978 | 0.994 | 0.993 | 0.988 | 0.985 | 0.977 | 0.995 |
| GraphRNN-100% | 0.675 | 0.806 | 0.822 | 0.579 | 0.974 | 0.802 | 0.495 | 0.583 | 0.643 | 0.715 |
| GraphRNN-66% | 0.607 | 0.834 | 0.9 | 0.624 | 0.99 | 0.799 | 0.596 | 0.74 | 0.545 | 0.792 |
| GRAN-100% | 0.614 | 0.818 | 0.885 | 0.578 | 0.879 | 0.714 | 0.737 | 0.66 | 0.524 | 0.781 |
| GRAN-66% | 0.672 | 0.805 | 0.89 | 0.615 | 0.969 | 0.818 | 0.669 | 0.679 | 0.675 | 0.76 |

(d) Using the F1 PR metric. Part B.

|  | 10 | 11 | 12 | 13 | 14 | 15 | 16 | 17 | 18 | 19 |
|---|---|---|---|---|---|---|---|---|---|---|
| 50/50 split | 0.989 | 0.993 | 0.991 | 0.99 | 0.996 | 0.992 | 0.993 | 0.983 | 0.986 | 0.99 |
| GraphRNN-100% | 0.979 | 0.724 | 0.638 | 0.552 | 0.751 | 0.71 | 0.69 | 0.813 | 0.708 | 0.768 |
| GraphRNN-66% | 0.983 | 0.697 | 0.749 | 0.619 | 0.9 | 0.695 | 0.691 | 0.864 | 0.62 | 0.772 |
| GRAN-100% | 0.932 | 0.719 | 0.615 | 0.597 | 0.925 | 0.682 | 0.667 | 0.788 | 0.703 | 0.719 |
| GRAN-66% | 0.956 | 0.705 | 0.679 | 0.653 | 0.895 | 0.717 | 0.679 | 0.879 | 0.749 | 0.774 |

Table 15: Evaluating GGMs at various stages of training on the **Grid** dataset using all 20 GIN architectures tested in the main-body.

(a) Using the MMD RBF metric. Part A.

|              | 0 | 1 | 2 | 3 | 4 | 5 | 6 | 7 | 8 | 9 |
|--------------|-------|-------|-------|-------|-------|-------|-------|-------|-------|-------|
| 50/50 split  | 0.043 | 0.05 | 0.051 | 0.047 | 0.043 | 0.047 | 0.054 | 0.043 | 0.043 | 0.041 |
| GraphRNN-100% | 0.195 | 0.206 | 0.238 | 0.202 | 0.211 | 0.175 | 0.149 | 0.215 | 0.199 | 0.174 |
| GraphRNN-66% | 0.155 | 0.18 | 0.222 | 0.17 | 0.221 | 0.154 | 0.183 | 0.166 | 0.175 | 0.242 |
| GRAN-100%    | 0.065 | 0.063 | 0.061 | 0.064 | 0.06 | 0.055 | 0.064 | 0.057 | 0.062 | 0.063 |
| GRAN-66%     | 0.06 | 0.06 | 0.059 | 0.06 | 0.059 | 0.053 | 0.061 | 0.055 | 0.059 | 0.06 |

(b) Using the MMD RBF metric. Part B

|              | 10 | 11 | 12 | 13 | 14 | 15 | 16 | 17 | 18 | 19 |
|--------------|-------|-------|-------|-------|-------|-------|-------|-------|-------|-------|
| 50/50 split  | 0.042 | 0.052 | 0.047 | 0.041 | 0.039 | 0.045 | 0.047 | 0.046 | 0.054 | 0.047 |
| GraphRNN-100% | 0.184 | 0.18 | 0.283 | 0.158 | 0.23 | 0.248 | 0.226 | 0.18 | 0.205 | 0.149 |
| GraphRNN-66% | 0.154 | 0.164 | 0.285 | 0.17 | 0.181 | 0.191 | 0.154 | 0.146 | 0.145 | 0.231 |
| GRAN-100%    | 0.063 | 0.065 | 0.06 | 0.063 | 0.062 | 0.073 | 0.063 | 0.062 | 0.065 | 0.064 |
| GRAN-66%     | 0.061 | 0.06 | 0.058 | 0.057 | 0.059 | 0.075 | 0.06 | 0.059 | 0.06 | 0.06 |

(c) Using the F1 PR metric. Part A.

|              | 0 | 1 | 2 | 3 | 4 | 5 | 6 | 7 | 8 | 9 |
|--------------|-------|-------|-------|-------|-------|-------|-------|-------|-------|-------|
| 50/50 split  | 0.995 | 0.996 | 0.997 | 0.998 | 0.997 | 0.997 | 0.997 | 0.998 | 0.997 | 1.0 |
| GraphRNN-100% | 0.955 | 0.964 | 0.969 | 0.947 | 0.958 | 0.958 | 0.919 | 0.95 | 0.95 | 0.935 |
| GraphRNN-66% | 0.953 | 0.915 | 0.934 | 0.969 | 0.964 | 0.955 | 0.913 | 0.93 | 0.964 | 0.969 |
| GRAN-100%    | 1.0 | 1.0 | 1.0 | 1.0 | 1.0 | 1.0 | 0.98 | 1.0 | 0.99 | 1.0 |
| GRAN-66%     | 0.995 | 0.995 | 1.0 | 1.0 | 1.0 | 1.0 | 0.953 | 1.0 | 0.985 | 1.0 |

(d) Using the F1 PR metric. Part B.

|              | 10 | 11 | 12 | 13 | 14 | 15 | 16 | 17 | 18 | 19 |
|--------------|-------|-------|-------|-------|-------|-------|-------|-------|-------|-------|
| 50/50 split  | 0.998 | 0.997 | 1.0 | 0.993 | 0.999 | 0.997 | 0.997 | 0.997 | 1.0 | 1.0 |
| GraphRNN-100% | 0.919 | 0.985 | 0.958 | 0.945 | 0.914 | 0.924 | 0.958 | 0.93 | 0.924 | 0.958 |
| GraphRNN-66% | 0.912 | 0.985 | 0.935 | 0.98 | 0.935 | 0.88 | 0.974 | 0.91 | 0.98 | 0.945 |
| GRAN-100%    | 1.0 | 1.0 | 1.0 | 1.0 | 1.0 | 0.99 | 1.0 | 1.0 | 1.0 | 1.0 |
| GRAN-66%     | 1.0 | 1.0 | 0.995 | 0.99 | 1.0 | 0.99 | 0.995 | 1.0 | 1.0 | 1.0 |

