# OpenReview forum: "On Evaluation Metrics for Graph Generative Models"
_ICLR.cc/2022/Conference — ICLR 2022 Poster_

### Official Review · Reviewer_XmrB · 2021-10-23

**Correctness:** 3
**Technical Novelty And Significance:** 1
**Empirical Novelty And Significance:** 2
**Recommendation:** 6
**Confidence:** 3

**Main Review:**

In general, I think this paper provides a systematic way of evaluating the metrics and they are reasonable. But this paper provides lots of information that distracts the readers away from the main conclusions.

Strengths:
1. Many factors have been considered. Except for fidelity, I think mode collapse and dropping are very important factors but are often neglected, and I am glad to see those factors being considered here.

2. I think the randomized GIN-based metrics have some novelty there and I am surprised to see they are more robust to mode collapse and dropping.

Weakness:
1. The NN-based metrics seem to be bound to the model GIN, so it's not sure if randomized GNNs are always robust to mode collapse and dropping compared to pretrained GNNs.

2. When evaluating the rank correlations, the authors also vary different architectures (e.g, number of layers). Why do we require the metric to be insensitive to the number of layers? Or in other words, in Fig.3, where does the variance come from? Does the variance come from architectural changes or other factors?

3. In Fig3, what is $\hat{\rho}(S_g, S_r)$?

4. I am wondering whether graphs of different degree distributions will affect Fig.3? The datasets provided are quite small. I am wondering if you create several kinds of synthetic graphs and see for each type of graphs if your conclusions still hold.

**Summary Of The Paper:**

This paper evaluates the effectiveness of different metrics for graph generative models from many perspectives. They thoroughly study the following factors: fidelity, diversity, sensitivity to node/edge features. They find that pre-existing GGM metrics fail to capture the diversity of data and find several random GIN-based metrics that are more expressive and have low computational costs.

**Summary Of The Review:**

I think the paper has evaluated different metrics extensively and they have provided abundant information. However, I feel a bit lost when I read this paper due to distractions from the details. I think authors should make their conclusions more clear before providing details. Besides, a few more concerns about this paper have been mentioned above.

---

> ### Author Response · Authors · 2021-11-20
> **Thank you for your review (1/2)**
>
> Thank you for your feedback and positive comments. We’ve uploaded a revision and highlighted all changes that were made in direct response to reviewer feedback in blue.
>
> > But this paper provides lots of information that distracts the readers away from the main conclusions.
>
> There is a lot that goes into evaluating evaluation metrics, and we tried to do it as objectively and concisely as possible. Our updated revision includes several clarifications in response to the reviews. If the reviewer has any specific suggestions on sections/information that can be trimmed or reorganized to make it easier to understand (that aren’t addressed below) we would be happy to take these into consideration.
>
> > Many factors have been considered. Except for fidelity, I think mode collapse and dropping are very important factors but are often neglected, and I am glad to see those factors being considered here.
>
> We agree that a metric’s ability to measure diversity is extremely important and yet is often overlooked. It is especially alarming that the baseline metrics have been used for years, yet they do not accurately measure the diversity of generated graphs.
>
> > The NN-based metrics seem to be bound to the model GIN
>
> We have added a comparison that considers random GCN [1], GraphSAGE [2], and GIN [3] without concatenation (suggested by reviewer Ru33). To keep this comparison concise, we focus on the F1 PR and MMD RBF metrics. We found that the performance of MMD RBF was fairly constant across GNNs, while F1 PR had slightly more fluctuation yet relatively strong performance. This table also addresses reviewer Ru33’s comments as well. We have tempered the discussion in Section 3 regarding the importance of GIN’s injectivity to line up with the results from this experiment, however, these results back up our claims that the overall GNN architecture is not crucial to the evaluation pipeline. We’ve included the table below, for an easier to read version please see Appendix C.6 page 25. Note that these experiments are ~90% complete, so the numbers may change a little bit in our final revision. Let us know if there is anything else you would like to see here.
>
> Table A.
>
> |           Metric          |      Fidelity      |      Diversity     | Fidelity & Diverstiy |  Node/Edge feats. | Sample eff. |
> |:-------------------------:|:------------------:|:------------------:|:--------------------:|:-----------------:|:-----------:|
> | F1 PR (GraphSAGE)         | $0.9000 \pm 0.004$ | $0.920 \pm 0.004$  | $0.910 \pm 0.003$    | $0.990 \pm 0.001$ | $7 \pm 0$   |
> | F1 PR (GCN)               | $0.870 \pm 0.005$  | $0.910 \pm 0.005$  | $0.890 \pm 0.003$    | $0.990 \pm 0.000$ | $7 \pm 0$   |
> | F1 PR (GIN, no concat.)   | $0.790 \pm 0.007$  | $0.920 \pm 0.004$  | $0.860 \pm 0.004$    | $0.980 \pm 0.004$ | $7 \pm 0$   |
> | F1 PR (GIN)               | $0.920 \pm 0.004$  | $0.920 \pm 0.003$  | $0.930 \pm 0.003$    | $0.990 \pm 0.000$ | $7 \pm 0$   |
> | MMD RBF (GraphSAGE)       | $0.950 \pm 0.003$  | $0.950 \pm 0.003$  | $0.950 \pm 0.002$    | $1.000 \pm 0.001$ | $23 \pm 2$  |
> | MMD RBF (GCN)             | $0.950 \pm 0.003$  | $0.950 \pm 0.003$  | $0.950 \pm 0.002$    | $1.000 \pm 0.002$ | $55 \pm 3$  |
> | MMD RBF (GIN, no concat.) | $0.960 \pm 0.003$  | $0.940 \pm 0.003$  | $0.950 \pm 0.002$    | $0.990 \pm 0.005$ | $60 \pm 3$  |
> | MMD RBF (GIN)             | $0.970 \pm 0.002$  | $0.950 \pm 0.003$  | $0.960 \pm 0.002$    | $1.000 \pm 0.001$ | $42 \pm 2$  |

---

> > ### Author Response · Authors · 2021-11-20
> > **Thank you for your review (2/2)**
> >
> > > When evaluating the rank correlations, the authors also vary different architectures (e.g, number of layers). Why do we require the metric to be insensitive to the number of layers? Or in other words, in Fig.3, where does the variance come from? Does the variance come from architectural changes or other factors?
> >
> > Unlike the image domain where Inception v3 is the default for extracting embeddings from samples in generative model evaluation, there is no strong pretrained classifier for the graph domain that we can use. Thus, we consider the problems of both 1) finding a strong GNN architecture for GGM evaluation, and 2) finding a strong metric for GGM evaluation. While we do not require any metric to be insensitive to the number of layers (any one strong architecture would suffice), it is an interesting result. Let us provide some additional clarification with respect to the individual sources of variance in Figure 3. For all metrics, variance is introduced in this figure by changing the random seed which modifies the pseudorandom aspect of the experiments (i.e. the order graphs are mixed in, which edges are rewired at each iteration, etc.). Also, for all metrics we aggregate the results across all of our datasets, which introduces additional variance. Finally, the GIN-based metrics have additional variance that originates from 1) varying model weights (this is impacted by the random seed), and 2) varying the model architecture (the results in this figure combine results for all GIN architectures we tested). This is explained more formally in the last sentence of the “Evaluating the evaluation metrics” paragraph in Section 4.
> >
> > > In Fig3, what is ρ^(Sg,Sr)?
> >
> > This is the mean value of each metric throughout the given experiment. We have made this more clear in the paper.
> >
> > > I am wondering whether graphs of different degree distributions will affect Fig.3? The datasets provided are quite small. I am wondering if you create several kinds of synthetic graphs and see for each type of graphs if your conclusions still hold.
> >
> > Figure 3 aggregates results across all of our datasets, which contains both synthetic (Grid, Lobster, Community), and real (Ego, Proteins) datasets. However, for a more thorough investigation we have added results for each dataset individually in Appendix C.3 pages 21 and 22. Unfortunately, these tables are a bit too large to include in OpenReview, but we encourage you to check it out. We find that there are no obvious outliers and it appears that all conclusions we draw still apply to each individual dataset. In regards to your comment about the size of each dataset — although many of the datasets are synthetic and we can generate an arbitrary number of graphs, we utilize the same datasets found in previous GGM papers for consistency. We mention this in the "Datasets" paragraph in Section 4. Please let us know if there’s anything else you would like to see here or if you have any other suggestions for improvement.

---

> > > ### Comment · Reviewer_XmrB · 2021-11-29
> > > **Feedback for the response**
> > >
> > > I appreciate the efforts of the authors to provide additional experiments using other models. Thus, I raise the score to 6.

---

### Official Review · Reviewer_UXK8 · 2021-10-24

**Correctness:** 3
**Technical Novelty And Significance:** 2
**Empirical Novelty And Significance:** 2
**Recommendation:** 6
**Confidence:** 4

**Details Of Ethics Concerns:**

None identified.

**Main Review:**

While I appreciate the general idea (that of using untrained/pretrained GNNs to extract graph embeddings and use these representations to compare sampled vs. real graphs), this paper has, in my opinion, two major defects (the second being much more important than the first).

1) The paper is confusing and makes some incorrect claims.
In the abstract, the authors say that they introduce _one_ scalar metric for graph evaluation, yet in the discussion, they recommend _two_. Moreover, the authors claim that current metrics ignore node and edge features. This is not true: e.g. in https://arxiv.org/abs/2001.08184 and https://arxiv.org/abs/2107.08396 the distributions of node and edge labels are compared. Unfortunately, from the reader's perspective, these two points give the impression that the authors are not sure about which metric to use and they did not check the literature very well.

2) Assuming that the proposed metric is random GIN + MMD RBF on the extracted graph representations, I argue this metric is not useful in the most important application field of GGMs: molecular generation. In fact, the chemical validity of a molecule cannot be captured by the graph representation in principle. I can think of an extreme example where the generated sample is composed of invalid molecules which only differ in one atom from the real sample. This generated sample would score close to 0 with the proposed metric, while in fact, it should be as close to 1 as possible. Of course, one could think of filtering out the invalid molecules first, but that defeats the purpose of having a unique metric in the first place. In my opinion (which I'll be happy to discuss), the world of graphs is not as regular as that of images; thus, the "one metric to rule them all" paradigm is not convincing to me.

**Summary Of The Paper:**

The paper proposes a scalar metric for evaluating Graph Generative Models (GGMs). The metric is based on computing the Maximum Mean Discrepancy (with an RBF kernel) between graph representations of the sampled and real graphs, as extracted from an untrained GIN model. In the paper, the authors analyze several metrics (some used in the literature, some similar to the proposed setup, e.g. replacing MMD), and measure their fidelity (i.e. how sensitive is the metric to random graphs), diversity (i.e. how sensitive it is to mode collapse or mode dropping), sensitivity to node/edge features, sample efficiency (minimum number of graphs necessary to discriminate noise from real samples), and computational efficiency.

**Summary Of The Review:**

I recommend rejection for now, for the motivations above. I will however await for the authors response and will update my claim accordingly.

EDIT 1
Raised my score to 5 after the first round of review and a first discussion with the authors.

EDIT 2
Raised my score to 6 considering the improved quality of the paper and the effort shown by the authors to fix what was wrong in the first version of the paper.

---

> ### Author Response · Authors · 2021-11-18
> **Thank you for your review! (1/2)**
>
> Thank you for your critique and openness to discussion, we look forward to hearing back from you.
>
> > In the abstract, the authors say that they introduce one scalar metric for graph evaluation, yet in the discussion, they recommend two.
>
> We will update the abstract to be consistent with our main message outlined in the paragraph after Table 3 in the paper (Section 5). To recap, our main message is a very simple “decision rule” that researchers can use to decide on which metric to use. More specifically, for a given dataset we recommend either MMD RBF or F1 PR depending on the quantity of samples in the dataset.
>
> > Moreover, the authors claim that current metrics ignore node and edge features. This is not true.
>
> We were aware of the NSPDK metric only through [1], which led us to (wrongly) believe it was infrequently used. In addition, [1] does not release open-source code which would have made it difficult for us to provide a fair comparison. However, [2] linked in this review, provides open-source code so we were able to experiment with this metric. Similar to Table 3, we summarize the results for this metric below:
>
> |   Metric  |      Fidelity     |     Diversity     | Fidelity & Diversity |  Node/Edge feats. | Sample eff. | Comp. eff. (s) |
> |:---------:|:-----------------:|:-----------------:|:--------------------:|:-----------------:|:-----------:|:----------------:|
> | NSPDK MMD | $0.990 \pm 0.002$ | $0.650 \pm 0.077$ | $0.810 \pm 0.044$    | $1.000 \pm 0.000$ | $8 \pm 1$   | 382            |
> | F1 PR     | $0.920 \pm 0.004$ | $0.930 \pm 0.003$ | $0.930 \pm 0.003$    | $0.990 \pm 0.000$ | $7 \pm 0$   | 18             |
> | MMD RBF   | $0.970 \pm 0.002$ | $0.950 \pm 0.003$ | $0.960 \pm 0.002$    | $1.000 \pm 0.001$ | $42 \pm 2$  | 120            |
>
> While this metric does well at incorporating node and edge features in evaluation, it is plagued by its poor ability to measure diversity of generated graphs. It is also incapable of incorporating continuous node and edge features [3] in evaluation while the GNN-based metrics have no such limitation. We’re working on incorporating the NSPDK metric into the main-body of our paper and tempering our original claim that previous metrics don’t consider node and edge features. These changes will be present by the author response deadline, however, we wanted to give you some time to review our progress and reply. Also, we estimate that the experiments evaluating the NSPDK metric are ~50% complete; the exact numbers will change slightly but the conclusions are unlikely to change. We will provide an update at 100%.

---

> > ### Author Response · Authors · 2021-11-18
> > **Thank you for your review! (2/2)**
> >
> > > I argue this metric is not useful in the most important application field of GGMs: molecular generation. In fact, the chemical validity of a molecule cannot be captured by the graph representation in principle. I can think of an extreme example where the generated sample is composed of invalid molecules which only differ in one atom from the real sample. This generated sample would score close to 0 with the proposed metric, while in fact, it should be as close to 1 as possible
> >
> > We think this critique is a result of a miscommunication on our part. We’re primarily interested in metrics that measure the distance between two distributions of graphs. As you say, the graph domain is not as regular as images and there are always going to be beneficial domain-specific metrics (e.g. % valid as you mentioned, drug-likeness is another), and there is no reasonable method to combine these into a scalar metric while remaining domain-agnostic. However, these domain-specific metrics are based on *properties* of the generated graphs and *do not provide a comparison to the reference distribution*. Our goal is not to eliminate these metrics, but rather to provide a strong domain-agnostic method for comparing generated graphs to a reference distribution that can be used in conjunction with these domain-specific metrics. With these two components evaluated using separate metrics, we’d definitely argue that in your example our metric should remain close to 0 as these graphs are still extremely similar. Again, we will make this point more clear in our paper by the deadline but we wanted to leave extra time for discussion. We are interested in hearing what you think regarding the points we’ve made and if you still stand by your original critique.
> >
> > > I argue this metric is not useful in the most important application field of GGMs: molecular generation.
> >
> > While molecular generation is certainly the most active area of GGM research, it is not the only area. Many papers [1, 2, 4-9] target unattributed (or non-molecular) graph generation in some capacity. All of these use the baseline metrics from You et. al. in evaluation *and* receive conflicting rankings across the three metrics. In addition, the papers you linked [2, 4] use a whopping 11 metrics (!) in evaluation. We strongly believe that the quantity of metrics in GGM papers needs to be narrowed in order to adequately measure progress.
> >
> > [1] Kawai, Wataru, et al. “Scalable Generative Models for Graphs with Graph Attention Mechanism.” 3 Oct. 2019, https://arxiv.org/abs/1906.01861.
> >
> > [2] Goyal, Nikhil, et al. “GraphGen: A Scalable Approach to Domain-Agnostic Labeled Graph Generation.” 8 Apr. 2020, https://arxiv.org/abs/2001.08184.
> >
> > [3] Costa, Fabrizio, and Kurt De Grave. “Fast Neighborhood Subgraph Pairwise Distance Kernel.” Proceedings of the 27th International Conference on International Conference on Machine Learning, 1 June 2010, https://dl.acm.org/doi/10.5555/3104322.3104356.
> >
> > [4] Podda, Marco, and Davide Bacciu. “GraphGen-Redux: a Fast and Lightweight Recurrent Model for Labeled Graph Generation.” 18 July 2021, https://arxiv.org/abs/2107.08396.
> >
> > [5] You, Jiaxuan, et al. “GraphRNN: Generating Realistic Graphs with Deep Auto-Regressive Models.” 23 June 2018, https://arxiv.org/abs/1802.08773.
> >
> > [6] Liao, Renjie, et al. “Efficient Graph Generation with Graph Recurrent Attention Networks.” 17 July 2020, https://arxiv.org/abs/1910.00760.
> >
> > [7] Shi, Chence, et al. “GraphAF: a Flow-Based Autoregressive Model for Molecular Graph Generation.” 27 Feb. 2020, https://arxiv.org/abs/2001.09382.
> >
> > [8] Niu, Chenhao, et al. “Permutation Invariant Graph Generation via Score-Based Generative Modeling.” 2 Mar. 2020, https://arxiv.org/abs/2003.00638.
> >
> > [9] Fan, Shuangfei, and Bert Huang. “Labeled Graph Generative Adversarial Networks.” 25 Feb. 2021, https://arxiv.org/abs/1906.03220.

---

> > > ### Comment · Reviewer_UXK8 · 2021-11-18
> > > **Thanks for the response**
> > >
> > > Thanks for your reply. Indeed, your answers are clarifying. I think I now understand your point of view better. Let me comment on your answer, so we can update our discussion.
> > >
> > > "We were aware of the NSPDK metric only through [1], which led us to (wrongly) believe it was infrequently used. In addition, [1] does not release open-source code which would have made it difficult for us to provide a fair comparison."
> > >
> > > To be clear: my point was not specifically related to NSPDK per se. My point is that there are metrics that measure some kind of distance between the real and generated distributions accounting for node and edge features, so your initial claim that there were none was not correct. I see you will update the paper to fix the claim, and I really appreciate that you performed an additional comparison to show that NSPDK is not good at quantifying diversity. This, in my opinion, adds value to your paper, providing another justification as to why one should use the proposed metric instead of those appearing in the literature. So, as regards this point, I am now completely in agreement with you, and I consider this issue solved.
> > >
> > > "Our goal is not to eliminate these metrics, but rather to provide a strong domain-agnostic method for comparing generated graphs to a reference distribution that can be used in conjunction with these domain-specific metrics."
> > >
> > > Okay, I get the point. I am sorry, but throughout the paper, I was left with the impression that the proposed metric was all that you need to evaluate GGMs. I now understand this is not the case and I get your intent. I also consider this issue solved, but I encourage you to make this concept as clear as possible in your revision.
> > >
> > >  "In addition, the papers you linked [2, 4] use a whopping 11 metrics (!) in evaluation. We strongly believe that the quantity of metrics in GGM papers needs to be narrowed in order to adequately measure progress."
> > >
> > > At the moment, this is the only point where I strongly disagree with you. I really cannot understand why that's so desperately needed. As an analogy, in standard classification, you would measure accuracy to measure the predictive ability, but also precision and recall to get an understanding of which kinds of errors the model is making. Similarly, and more so in the graphs domain, where there are lots of things to account for (nodes, edges, communities, disconnected components, and whatnot), I don't find it so strange to have a set of metrics that quantify different aspects of the generated graphs, rather than just one. Then, depending on the specific problem, one can base its qualitative judgement on one single "business metric" which pinpoints the aspect one is interested to preserve the most during the generation. I'm just failing to see why having one general instead of many specific is so detrimental to advance in this field.
> > >
> > > After your reply, I'm raising the paper's score to a 5 to show my appreciation of your efforts to improve its quality. Having read the concerns of the other reviewers, I agree with them that the paper is borderline. For the moment being, I am more inclined towards rejection, but I will update again my final score after reading the other discussions.

---

> > > > ### Author Response · Authors · 2021-11-20
> > > > **Thank you for your reply (1/2)**
> > > >
> > > > First of all, we would like to thank you for your very quick reply and again your openness to discussion. It is much appreciated. We are happy to hear that you appreciated our clarification and changes that we’re making. We have uploaded a revision addressing the comments of all reviewers and encourage you to check it out. Changes that have been made in direct response to reviewer feedback is coloured in blue. Regarding the previous clarity issue that our metrics are all that is needed to evaluate GGMs, we have updated the abstract, added detail to a sentence in the introduction, a couple of sentences at the end of our introduction to classical metrics (Section 2.1, top of page 3), and a footnote in our discussion (page 9). Please let us know if you feel we are still unclear or if there are any improvements to be made.
> > > >
> > > > Regarding the last sticky point, we actually feel that we are close to being on the same page. We agree, similar to accuracy/precision/recall (a/p/r), having specific metrics that measure certain properties can be beneficial in certain scenarios. If a researcher is interested in generating graphs with a similar degree distribution to a training set, by all means performing model selection and measuring progress using Degree MMD would be a great idea. However, our point is that this is *not* how these metrics are typically used. Researchers typically display all metrics together to approximate the quality of generated graphs. They rarely (if ever) focus on a *standalone* metric (or “business metric”). As the goal is usually to generate graphs resembling a reference distribution and it is *unclear which of these metrics measures this the best*, GGM papers that use these metrics present results with the general goal of reducing all of Deg./Clus./Orbits MMD simultaneously. Thus, if the goal is to generally resemble a reference distribution (as in [1-9]), it is unclear which metric should be chosen.  We think the presentation of results by [1] (Table 2, page 8) highlights this point: they take the average across all three metrics to determine which model is superior. Unlike a/p/r where each metric measures something very specific, here all three metrics are used to attempt to measure the quality of generated graphs. This causes issues when the metrics do not agree on which model should be selected, or whether the approach of one paper is superior to another. This is one of the things we tried to highlight with our section on GGM selection and Table 2 (page 9): when all of these metrics are used to measure the same thing, inconsistencies can and do arise. By design, condensing these into a strong single-value metric (as [1] attempted to do with a simple average) allows progress to be measured more reliably. Identifying one or more standalone metrics (“business metrics”) that can be used in this scenario is really the problem we try to tackle, and upon review we understand how this was unclear in our original manuscript. In the recent revision we have tried to make this more clear. We have added a sentence to the introduction that adds detail surrounding how these metrics are typically used, and a sentence to our discussion that there are still specific use-cases for such metrics. We have also clarified the goal of Table 2. Please let us know if you still feel like this is an issue or if we can be more clear somewhere.

---

> > > > > ### Author Response · Authors · 2021-11-20
> > > > > **Thank you for your reply (2/2)**
> > > > >
> > > > > While we feel we have made our point clear, here are a few other examples that support our argument:
> > > > >
> > > > > From [1] again (page 6): “*To evaluate generation quality*, we used maximum mean discrepancy (MMD) over some graph statistics, as proposed by You et al. (2018b). We calculated MMD for three graph statistics: 1) degree distribution, 2) cluster coefficient distribution, and 3) the number of orbits with 4 nodes.”
> > > > >
> > > > > From [2] (page 5): “*To evaluate the quality of the generated graphs*, we follow the approach used by You et al. (2018a): ... We use four graph statistics to evaluate the generated graphs: degree distribution, clustering coefficient distribution, node-label distribution, and average orbit count statistics.”
> > > > >
> > > > > From [3] (page 7): “*To evaluate the quality of generation*, evaluation metrics based on MMD with graph statistics were used [as] in You et al. (2018b).”
> > > > >
> > > > > From [4] (page 8): “*For evaluation*, we report the Maximum Mean Discrepancy (MMD) (Gretton et al., 2012) between generated and training graphs using some specific metrics on graphs proposed by You et al. (2018b)”
> > > > >
> > > > > From [5] (page 7): “... we follow [37, 21] and *evaluate model performance* by comparing the distributions of graph statistics between the generated and ground truth graphs”
> > > > >
> > > > > From [6] (page 7): “We use three graph statistics—based on degrees, clustering coefficients and orbit counts—to further quantitatively *evaluate the generated graphs*.”
> > > > >
> > > > > [1] Niu, Chenhao, et al. “Permutation Invariant Graph Generation via Score-Based Generative Modeling.” 2 Mar. 2020, https://arxiv.org/abs/2003.00638.
> > > > >
> > > > > [2] Fan, Shuangfei, and Bert Huang. “Labeled Graph Generative Adversarial Networks.” 25 Feb. 2021, https://arxiv.org/abs/1906.03220.
> > > > >
> > > > > [3] Kawai, Wataru, et al. “Scalable Generative Models for Graphs with Graph Attention Mechanism.” 3 Oct. 2019, https://arxiv.org/abs/1906.01861.
> > > > >
> > > > > [4] Shi, Chence, et al. “GraphAF: a Flow-Based Autoregressive Model for Molecular Graph Generation.” 27 Feb. 2020, https://arxiv.org/abs/2001.09382.
> > > > >
> > > > > [5] Liao, Renjie, et al. “Efficient Graph Generation with Graph Recurrent Attention Networks.” 17 July 2020, https://arxiv.org/abs/1910.00760.
> > > > >
> > > > > [6] You, Jiaxuan, et al. “GraphRNN: Generating Realistic Graphs with Deep Auto-Regressive Models.” 23 June 2018, https://arxiv.org/abs/1802.08773.
> > > > >
> > > > > [7] Kawai, Wataru, et al. “Scalable Generative Models for Graphs with Graph Attention Mechanism.” 3 Oct. 2019, https://arxiv.org/abs/1906.01861.
> > > > >
> > > > > [8] Goyal, Nikhil, et al. “GraphGen: A Scalable Approach to Domain-Agnostic Labeled Graph Generation.” 8 Apr. 2020, https://arxiv.org/abs/2001.08184.
> > > > >
> > > > > [9] Podda, Marco, and Davide Bacciu. “GraphGen-Redux: a Fast and Lightweight Recurrent Model for Labeled Graph Generation.” 18 July 2021, https://arxiv.org/abs/2107.08396.

---

> > > > > ### Comment · Reviewer_UXK8 · 2021-11-22
> > > > > **Now it's clearer, and another question**
> > > > >
> > > > > I read your revised version, and the scope of the paper does indeed seem clearer now. I really appreciate your effort. When I was reading the supplementary, I stumbled on table 4 and 5 and it got me thinking. Imagine that I obtain a certain MMD with a fixed configuration, which isn't better than a competitor. I imagine I could "hack" my performance by finding a GIN configuration for which my results are better than the ones I'm comparing to. How would you deal with such a scenario? Would you simply take the mean across different configurations (and doesn't this overcomplicates things)? It seems to me that the fact that the GIN architecture can vary creates a bit of "fuzzyness" around this metric, which could be maliciously exploited.

---

> > > > > > ### Author Response · Authors · 2021-11-24
> > > > > > **Reply to question**
> > > > > >
> > > > > > We attempt to mitigate this issue by suggesting a single GIN architecture (3 propagation rounds, node embedding size of 35) to future researchers in Section 5 (page 9) as we show it to work well across a variety of datasets. To provide a fair comparison, researchers should be using this recommendation or providing strong justification for altering it (i.e. running our experiments and showing our suggested configuration is insufficient for their dataset).  In terms of the variance within this single architecture induced by changing model parameterizations (i.e. random inits): this is the second thing we try to address with Table 2. We show that this variance for MMD RBF (and others in the Appendix) is relatively negligible. We feel that we have made this more clear in our revision. Finally, we note that the GIN architecture is really just a parameter of the metrics we introduce, and having tweakable parameters in metrics is common. For example, O’Bray et al. (2021) [1] show that you can artificially manipulate the ranking of GGMs by tweaking the parameters of the graph statistic metrics from You et al. (2018) [2]. Nevertheless, investigating this in regards to our work is a worthwhile experiment. We have added Appendix C.9 (pages 27-29) (in our second most recent revision, https://openreview.net/references/pdf?id=snLqWyecpB) wherein we investigate the consistency of GGM rankings across GIN configurations. We find that artificially manipulating the ranking of GGMs by tweaking the GIN configuration is more difficult (within our space of 20 GIN configurations) than it sounds for the Grid and Proteins datasets. Based on the raw rankings, Lobster appears to be slightly more inconsistent. However, we note that the GGMs for this dataset are all extremely competitive with each other across all metrics (Table 12c, page 26) and the sample sizes are extremely small meaning simply looking at the raw rankings does not tell the full story. Finally, the architecture we propose is #10 in these tables, and it never appears to be an outlier relative to the others.
> > > > > >
> > > > > > [1] O'Bray, Leslie, et al. “Evaluation Metrics for Graph Generative Models: Problems, Pitfalls, and Practical Solutions.” 11 Oct. 2021, https://arxiv.org/abs/2106.01098.
> > > > > >
> > > > > > [2] You, Jiaxuan, et al. “GraphRNN: Generating Realistic Graphs with Deep Auto-Regressive Models.” 23 June 2018, https://arxiv.org/abs/1802.08773.

---

> > > > > > > ### Comment · Reviewer_UXK8 · 2021-11-24
> > > > > > > **Thanks**
> > > > > > >
> > > > > > > Alright, that answers my question in a satisfactory manner. I raised my score to 6 in light of your openness to discussion and to the effort you put into improving the original version. It's still a borderline paper, but after our interaction I now think it's on the good side of the border. Thanks!

---

> > > > > > > > ### Author Response · Authors · 2021-11-29
> > > > > > > > **Thanks for the discussion**
> > > > > > > >
> > > > > > > > Thanks for the discussion and helping to improve the paper, it is much appreciated!

---

### Official Review · Reviewer_CNrk · 2021-11-02

**Correctness:** 3
**Technical Novelty And Significance:** 2
**Empirical Novelty And Significance:** 2
**Recommendation:** 6
**Confidence:** 4

**Main Review:**

The inclusion of a new method/framework for GGM evaluation is always welcome in the literature. The paper points out the main deficiencies and creates a new metric to evaluate GGM.

One advantage of this method is its simplicity. As it is explained in the paper, a GIN is applied and then the evaluation is realized. Unfortunately, this simplicity came with a cost, there is no theoretical contribution. It will be interesting to understand from a theoretical point of view, why this metric can actually solve the problems mentioned by the authors, instead of showing everything empirically. For example, there is no analysis of the time complexity.

There is also a phrase that avoids previous advances of the GGM. "Frequently used datasets in the GGM literature are relatively small in size". While this phrase could be true for GGM based on GNN, there are multiple GGM (Chung-Lu, mKPGM, BTER, etc) that are able to replicate networks with millions of nodes and edges, sampling them with a time complexity proportional to the number of edges (networks with millions of nodes and edges are sampled in minutes).

Besides, even though the metric is able to solve some of the problems previously described, given the use of a GGN the embedding is impossible to interpret (neither its evaluation). For example, what does it mean a value of 0.97 in some of the metrics?

The comparison is against MMD which has several flaws, but the KS-multidimensional solves the same limitations mentioned in the paper. Moreover, the final value of the KS-multidimensional metric is also interpretable.

**Summary Of The Paper:**

The paper proposes the use of an untrained graph neural network (GNN) to generate a graph embedding which is used with other measures to evaluate Generative Graph Models (GGMs). The main advantages of this evaluation process are the use of a single score, the inclusion of node and edge features, and its empirical time complexity.

**Summary Of The Review:**

While the presentation of a new method/framework is important for the GGM, the paper lacks theory, and the new metric is impossible to interpret. Moreover, the three critical points mentioned in the paper have already been solved by the KS-multidimensional distance.

---

> ### Author Response · Authors · 2021-11-20
> **Thank you for your review**
>
> Thank you for your thoughtful review, we are glad to hear that you agree on the importance of GGM evaluation. We’ve uploaded a revision and highlighted all changes that were made in direct response to reviewer feedback in blue.
>
> > It will be interesting to understand from a theoretical point of view, why this metric can actually solve the problems mentioned by the authors, instead of showing everything empirically.
>
> We acknowledge the value of theoretical works such as [1] that can be used to prove the preservation of distances via random GIN. However, the main goal of our paper was to provide an empirical comparison of the metrics and yield insights that researchers can directly apply in future work. Therefore we feel a strong theoretical justification is out of scope.
>
> > There is also a phrase that avoids previous advances of the GGM. "Frequently used datasets in the GGM literature are relatively small in size".
>
> We did not mean this statement to discount the contributions of the non-GNN-based methods such as those that you listed. Rather, our point was that researchers occasionally (or frequently in the case of GNN-based generative models) use small datasets in GGM papers. Because of this, we need to investigate the sample efficiency of each metric thoroughly as we cannot just abstract away the problem and rely on the law of large numbers to solve it. We have rephrased this sentence as “The sample efficiency of each metric is extremely important as small datasets are frequently used in the GGM literature.”
>
> > Besides, even though the metric is able to solve some of the problems previously described, given the use of a GGN the embedding is impossible to interpret (neither its evaluation). For example, what does it mean a value of 0.97 in some of the metrics?
>
> While the metrics we propose do indeed lack interpretability, we argue that this isn’t a major issue (and if it is, it is a problem that also plagues our baseline metrics). Almost all of our metrics are adapted from the image generation domain which has seen rapid progress in recent years on the backs of network-based metrics. Also, when it comes to actually evaluating GGMs, we always computed each metric using a 50/50 split of the dataset. This provides information regarding what score two sets of indistinguishable graphs may receive and represents the ideal value for the given dataset. By comparing this value to the score of a model, it is easier to determine how much better or worse an alternative model is, and also provides information regarding how far away they are from an ideal value. We’ve added a comment to this section that future work should follow a similar process to improve interpretability.
>
> > The comparison is against MMD which has several flaws, but the KS-multidimensional solves the same limitations mentioned in the paper. ... Moreover, the three critical points mentioned in the paper have already been solved by the KS-multidimensional distance.
>
> We believe the reviewer is referring to the KS metric used in the mKPGM paper mentioned [2]. This assumption is what our reply is based on, however, if there is another paper that uses a separate KS metric that we’ve missed please let us know. Thank you for the heads up regarding this, we’ve added a mention to it in our background section, however, as it is not frequently used in GGM evaluation we’ve excluded it from our experiments for now. As it stands in the mKPGM paper, this metric is similar to the baseline metrics in that it requires determining the degree and clustering coefficients of each node, as well as the average geodesic distance of each node. While using KS-distance with these properties is a nice way of combining them into a single score, we see a few issues with this approach. We would expect it to suffer from the same computational efficiency problems as the baseline metrics as it extracts (in some cases) identical features from each graph. In addition, none of degree, clustering coefficient, or geodesic distance consider node and edge features, meaning it does not satisfy this critical point.
>
> [1] Zambon, Daniele, et al. “Graph Random Neural Features for Distance-Preserving Graph Representations.” 2 June 2020, https://arxiv.org/abs/1909.03790.
>
> [2] Monero S. et al., Tied Kronecker Product Graph Models to Capture Variance in Network Populations, 2018. https://dl.acm.org/doi/pdf/10.1145/3161885

---

### Official Review · Reviewer_Ru33 · 2021-11-03

**Correctness:** 4
**Technical Novelty And Significance:** 3
**Empirical Novelty And Significance:** 3
**Recommendation:** 8
**Confidence:** 3

**Main Review:**

The paper is well written and the claims are well backed with experimental
evidence. As it shows strong analogies to previous works in how the evaluation
metrics were evaluated and which aspects of the evaluation it focuses on, the
technical novelty of this study is limited to the suggestion of using random
GNNs for extracting features from generated graphs and the evaluation of sample
efficiency. The empirical novelty is high (assuming other works on the
evaluation of graph generative models are considered concurrent) as it is
important to show the community the pitfalls of current GGM evaluation
approaches in order to ensure that research progresses in an unbiased manner.

I did find some aspects of the paper that could see some improvements though:
 - An exploration on the suitability of different GNNs would be extremely
   useful to the community. Particularly, interesting would be how important
   the concatenation of the layer readouts is. Here I imagine this to be of
   high necessity for the approach to work one would otherwise expect
   oversmoothing problems (especially for randomly initialized NNs).
 - The evaluation with regard to graph distribution perturbations only
   considers a single type of graph perturbation -- mixing with random ER
   graphs and edges being rewired to represent ER graphs. This gives rise to
   the question if the model considered (GIN with sum aggregation) is biased
   towards recognizing these types of distribution shift. I could imagine that
   a randomly initialized GIN computes something similar to higher order node
   degree statistics, which would be particularly suited for this task.
 - The parameter selection for MMD might have some complications to my
   understanding: Here the scale parameter of the RBF kernel is selected to
   maximally differentiate the two distributions, yet this criterion does not
   include to which degree samples that are actually from the same distribution
   get lower discrepancy compared to the samples which are OOD. While there is
   some evidence that this should statistically not be the case for
   characteristic kernels, these statistics would require a large sample size
   for this to be true with very high probability which is not the case in all
   the experiments.
 - Finally, I wonder how fair the comparison between the static graph
   featurizations and the newly proposed approach is. The static approaches
   using graph statistics rely on a non-parametric earth movers distance kernel
   and thus do not undergo any additional selection of the hyperparameters
   compared to the proposed approach of the paper. This is especially critical
   in Table 2 where the approaches are compared directly using their MMD values
   and rankings. I am not sure if this difference would manifest in changes to
   the rank correlations in Figure 3 though.


**Summary Of The Paper:**

The paper shows a detailed comparison of different graph generative model
evaluation metrics and highlights that current approaches for the evaluation of
GGMs are insufficient and perform poorly in terms of assessing diversity and
fidelity of generated samples. The paper pinpoints these issues to the reliance
of many metrics on a predefined set of features extracted from the generated
graphs. It claims that the efficacy of prior approaches is typically dependent
on the graph featurization and that this leads to inconsistencies in the
ranking of models when using different feature representations. It proposes to
instead use features from randomly initialized Graph Neural Networks as a basis
for the analysis and shows that this represents a competitive evaluation
approach.

**Summary Of The Review:**

The paper is well written and contributes some new thoughts and approaches for
the evaluation of GGMs in an unbiased manner.  I solely have minor
considerations with regard to the types of graph perturbations selected and the
selection of the MMD hyperparameters. Further, I question whether the
comparison classical evaluation approaches to the GNN based approaches is
completely fair.

---

> ### Author Response · Authors · 2021-11-20
> **Thank you for your review (1/2)**
>
> Thank you for your thoughtful review and feedback. We are happy that you found the paper well written and that our experiments support all of our claims. We’ve uploaded a revision and highlighted all changes that were made in direct response to reviewer feedback in blue. In regards to your specific feedback:
>
> > An exploration on the suitability of different GNNs would be extremely useful to the community.
>
> We have added a comparison that considers random GCN [1], GraphSAGE [2], and GIN [3] without concatenation as you suggested. To keep this comparison concise, we focus on the F1 PR and MMD RBF metrics. We found that the performance of MMD RBF was fairly constant across GNNs, while F1 PR had slightly more fluctuation yet relatively strong performance. This table also addresses reviewer XmrB’s comments as well. We have tempered the discussion in Section 3 regarding the importance of GIN’s injectivity to line up with the results from this experiment, however, these results back up our claims that the overall GNN architecture is not crucial to the evaluation pipeline. We’ve included the table below, for an easier to read version please see Appendix C.6 page 25. Note that these experiments are ~90% complete, so the numbers may change a little bit in our final revision. Let us know if there is anything else you would like to see here.
>
> Table A.
>
> |           Metric          |      Fidelity      |      Diversity     | Fidelity & Diversity |  Node/Edge feats. | Sample eff. |
> |:-------------------------:|:------------------:|:------------------:|:--------------------:|:-----------------:|:-----------:|
> | F1 PR (GraphSAGE)         | $0.9000 \pm 0.004$ | $0.920 \pm 0.004$  | $0.910 \pm 0.003$    | $0.990 \pm 0.001$ | $7 \pm 0$   |
> | F1 PR (GCN)               | $0.870 \pm 0.005$  | $0.910 \pm 0.005$  | $0.890 \pm 0.003$    | $0.990 \pm 0.000$ | $7 \pm 0$   |
> | F1 PR (GIN, no concat.)   | $0.790 \pm 0.007$  | $0.920 \pm 0.004$  | $0.860 \pm 0.004$    | $0.980 \pm 0.004$ | $7 \pm 0$   |
> | F1 PR (GIN)               | $0.920 \pm 0.004$  | $0.920 \pm 0.003$  | $0.930 \pm 0.003$    | $0.990 \pm 0.000$ | $7 \pm 0$   |
> | MMD RBF (GraphSAGE)       | $0.950 \pm 0.003$  | $0.950 \pm 0.003$  | $0.950 \pm 0.002$    | $1.000 \pm 0.001$ | $23 \pm 2$  |
> | MMD RBF (GCN)             | $0.950 \pm 0.003$  | $0.950 \pm 0.003$  | $0.950 \pm 0.002$    | $1.000 \pm 0.002$ | $55 \pm 3$  |
> | MMD RBF (GIN, no concat.) | $0.960 \pm 0.003$  | $0.940 \pm 0.003$  | $0.950 \pm 0.002$    | $0.990 \pm 0.005$ | $60 \pm 3$  |
> | MMD RBF (GIN)             | $0.970 \pm 0.002$  | $0.950 \pm 0.003$  | $0.960 \pm 0.002$    | $1.000 \pm 0.001$ | $42 \pm 2$  |
>
>
> > The evaluation with regard to graph distribution perturbations only considers a single type of graph perturbation -- mixing with random ER graphs and edges being rewired to represent ER graphs. This gives rise to the question if the model considered (GIN with sum aggregation) is biased towards recognizing these types of distribution shift.
>
> We added an experiment that mixes generated graphs from GRAN [4] as opposed to E-R graphs. Although this is a more difficult task, we found the performance of the metrics to be relatively consistent across these two experiments. We’ve included a table for F1 PR and MMD RBF below, please see Appendix C.4 page 23 for results for all metrics.
>
> Table B.
>
> |  Metric |   Mixing Random   |  Mixing Generated  |
> |:-------:|:-----------------:|:------------------:|
> | F1 PR   | $0.960 \pm 0.006$ | $0.970 \pm 0.005$  |
> | MMD RBF | $1.000 \pm 0.001$ | $0.990 \pm 0.002$  |

---

> > ### Author Response · Authors · 2021-11-20
> > **Thank you for your review (2/2)**
> >
> > > The parameter selection for MMD might have some complications to my understanding:
> >
> > We admit we may have misunderstood this comment but will do our best to address it. We think this might be suggesting the use of an adaptive (i.e. per example) sigma in our MMD implementation. If this is the case, we definitely believe that more complex approaches for selecting sigma would be interesting to tackle in future work. If this is the incorrect interpretation of your comment, please let us know and we will address it again.
> >
> > > The static approaches using graph statistics rely on a non-parametric earth movers distance kernel and thus do not undergo any additional selection of the hyperparameters compared to the proposed approach of the paper
> >
> > This was actually a typo on our part. The baseline metrics from You et. al. utilize the RBF kernel, however, they replace the L2-norm with the Earth Mover’s Distance (EMD) function. Thus, they have the same $\sigma$ parameter that our MMD RBF metric has, and throughout all of our experiments we simply used the values suggested by You et. al. We’ve corrected this mistake in the paper. While we can apply our $\sigma$ selection process to these metrics as well, we did not experiment with it as they would still be extremely expensive to compute and unable to incorporate node or edge features. However, we’ve added a comparison to the MMD RBF metric using a static $\sigma = 1$. We find that while it is weaker than the more complex version, it is still more expressive than the baseline metrics. This metric is now present in Appendix C.7 page 25, and we summarize the results below.
> >
> > Table C.
> >
> > |      Metric      |      Fidelity     |     Diversity     | Fidelity & Diversity | Node/edge feats. | Sample eff.  |
> > |:----------------:|:-----------------:|:-----------------:|:----------------------:|:------------------:|:--------------:|
> > | Orbits MMD       | $0.37 \pm 0.048$  | $0.49 \pm 0.046$  | $0.43 \pm 0.034$     |   N/A          | $122 \pm 22$ |
> > | Degree MMD       | $1.00 \pm 0.000$  | $0.51 \pm 0.061$  | $0.76 \pm 0.035$     |      N/A         | $9 \pm 1$    |
> > | Clustering MMD   | $0.99 \pm 0.003$  | $0.43 \pm 0.047$  | $0.72 \pm 0.030$     |        N/A       | $7 \pm 0$    |
> > | MMD RBF (static) | $0.97 \pm 0.002$  | $0.85 \pm 0.007$ | $0.91 \pm 0.004$      | $0.89 \pm 0.009$ | $33 \pm 2$   |
> > | MMD RBF          | $0.97 \pm 0.002$  | $0.95 \pm 0.003$  | $0.96 \pm 0.002$     | $1.00 \pm 0.001$ | $42 \pm 2$   |
> >
> > > This is especially critical in Table 2 where the approaches are compared directly using their MMD values and rankings.
> >
> > Our intention with Table 2 was not to provide a comparison of the raw MMD values (or rankings) between each approach. As the metrics are the best proxy we have for the quality of generated graphs, we intentionally avoid suggesting that there should be a certain ranking of the models presented. Training is noisy, and the datasets presented are simple enough that we don’t feel that a relatively early generative model checkpoint can’t compete with a later one. Rather, we use this table to try to highlight that 1) evaluation using the baseline metrics can and does produce inconsistent rankings. By design, using a single metric such as MMD RBF or F1 PR that captures all important properties eliminates this problem, and 2) that MMD RBF is low variance across random initializations. This is an attractive property as we don’t have to identify a good weight initialization. It’s also convenient as it means future researchers can simply follow our weight initialization process and we avoid any hassle of distributing trained models for evaluation. We have clarified this section of the paper as well as stated our goal with Table 2.
> >
> > [1] Kipf, Thomas N., and Max Welling. “Semi-Supervised Classification with Graph Convolutional Networks.” 22 Feb. 2017, https://arxiv.org/abs/1609.02907.
> >
> > [2] Hamilton, William L., et al. “Inductive Representation Learning on Large Graphs.” 10 Sept. 2018, https://arxiv.org/abs/1706.02216.
> >
> > [3] Xu, Keyulu, et al. “How Powerful Are Graph Neural Networks?” 22 Feb. 2019, https://arxiv.org/abs/1810.00826.
> >
> > [4] Liao, Renjie, et al. “Efficient Graph Generation with Graph Recurrent Attention Networks.” 17 July 2020, https://arxiv.org/abs/1910.00760.

---

> > > ### Comment · Reviewer_Ru33 · 2021-11-29
> > > **Thank you for the detailed response**
> > >
> > > Thanks for the detailed response, most of my concerns where addressed and clarified.
> > >
> > > I still have one concern with the paper related to the hyperparameter selection procedure of MMD for the baseline methods.
> > > I don't think that using MMD with a constant value for the lengthscale hyperparameter is an appropriate comparison partner (see the issues that arise for different values of $\sigma$ in O’Bray et al., 2021).
> > > As the authors themselves have indicated, selecting $\sigma$ inappropriately leads to a deterioration in performance compared to the "optimal" $\sigma$ selection. This is also the case for the comparison approaches.
> > >
> > > While I totally understand that MMD with earth mover distance is hard to scale, it is essential in order to obtain a fair comparison between the conventional approaches and the approach presented in the paper. Alternatively, the authors could run these experiments using a more scalable kernel such as the conventional RBF kernel. This would at least allow to be sure that the added effect is not solely due to the MMD hyperparameter selection procedure. I strongly encourage the authors to include this into the final version of the paper.
> > >
> > > Independently, the approach proposed has its merits and I don't believe that the above request will significantly impact them.
> > > Further, due to my late response the authors will be unable to actually answer my above request, such that I cannot hold my concerns against them in this case.
> > >
> > > I will increase my score to an accept in order to reflect that most of my concerns where addressed. I still strongly encourage the authors to include the above experiments / adapt the experiments such that all methods follow the same hyperparameter selection procedure.

---

> > > > ### Author Response · Authors · 2021-11-29
> > > > **Thanks for the reply**
> > > >
> > > > Thanks for the reply and for reviewing our response. On the outstanding question regarding hyperparameter selection for MMD, we are happy to run the suggested experiments ahead of the final revision.

---

### Author Response · Authors · 2021-11-23
**Response to all reviewers**

We would like to thank all reviewers for their time and thoughtful feedback. We believe the clarity of our writing and strength of our paper has been improved based on your feedback. However, it seems that the main goal of our work has not been communicated well and we would like to provide some clarification.

In recent years, evaluation of graph generative models (GGMs) has frequently been done through the graph statistic metrics (Deg., Clus., Orbits MMD) introduced by You et al. (2018) [6] (see for example [1-8]). As the goal is usually to generate graphs resembling a reference distribution, and it is unclear which of these metrics is the best at measuring this dissimilarity, *all* metrics are used in evaluation, and *all* metrics are used as a measure of generation quality. As each metric has the same interpretation (i.e. a measure of generation quality), issues can and do arise when the rankings of GGMs are inconsistent across metrics. This is one of the things we tried to highlight with Table 2 (page 9), however, you can also see examples of inconsistent rankings across these metrics in [1-8]. In addition, the computation of a *single* one of these metrics can be extremely expensive (which we show empirically), and they are unable to incorporate node and edge features in evaluation. To further complicate things, recent works have introduced additional metrics that are also intended to measure the quality of generated graphs [5, 7]. As you can imagine, having this many metrics (that all attempt to measure the same thing) makes it *extremely challenging* to measure progress in this field. By design, condensing these into strong single-value metrics allows progress to be measured more reliably. This problem of identifying *standalone* metrics that accurately measure the dissimilarity between two sets of graphs is really what we tackle in this paper, and we understand how this was unclear in our original manuscript. Finally, to clarify our final recommendation: depending on the quantity of samples, we recommend the use of either MMD RBF *or* F1 PR. This means the use cases for each are disjoint and we recommend a single metric for a given dataset, thereby solving the problem caused by an excessive number of evaluation metrics.

**We feel that all of the points mentioned above have been made more clear in our revision, and we have also addressed all concerns mentioned by reviewers**. We invite all reviewers to take another look at our work.

[1] Niu, Chenhao, et al. “Permutation Invariant Graph Generation via Score-Based Generative Modeling.” 2 Mar. 2020, https://arxiv.org/abs/2003.00638.

[2] Fan, Shuangfei, and Bert Huang. “Labeled Graph Generative Adversarial Networks.” 25 Feb. 2021, https://arxiv.org/abs/1906.03220.

[3] Kawai, Wataru, et al. “Scalable Generative Models for Graphs with Graph Attention Mechanism.” 3 Oct. 2019, https://arxiv.org/abs/1906.01861.

[4] Shi, Chence, et al. “GraphAF: a Flow-Based Autoregressive Model for Molecular Graph Generation.” 27 Feb. 2020, https://arxiv.org/abs/2001.09382.

[5] Liao, Renjie, et al. “Efficient Graph Generation with Graph Recurrent Attention Networks.” 17 July 2020, https://arxiv.org/abs/1910.00760.

[6] You, Jiaxuan, et al. “GraphRNN: Generating Realistic Graphs with Deep Auto-Regressive Models.” 23 June 2018, https://arxiv.org/abs/1802.08773.

[7] Goyal, Nikhil, et al. “GraphGen: A Scalable Approach to Domain-Agnostic Labeled Graph Generation.” 8 Apr. 2020, https://arxiv.org/abs/2001.08184.

[8] Podda, Marco, and Davide Bacciu. “GraphGen-Redux: a Fast and Lightweight Recurrent Model for Labeled Graph Generation.” 18 July 2021, https://arxiv.org/abs/2107.08396.

---

### Author Response · Authors · 2021-11-29
**Additional comments or concerns**

We'd like to politely request you to review our replies ahead of the Nov 29th deadline if you haven't already. If you have any additional comments or concerns we are happy to address them.

---

### Public Comment · ~Xiaohui_Chen2 · 2022-01-29
**Related works to graph generative model**

Hi, Congrats on your paper acceptance! I would like to introduce our work which is also related to graph generation and your work: https://arxiv.org/abs/2106.06189

---

> ### Public Comment · ~Rylee_Thompson1 · 2022-03-11
> **Thanks**
>
> Very cool work, thanks for bringing this up! We've added reference to your work in our local doc which we'll upload to OpenReview and arXiv in the coming days.

---

### Decision · Program_Chairs · 2022-01-20

**Decision:**

Accept (Poster)

**Comment:**

The paper argues that existing evaluation metrics for GGMs are insufficient and perform an extensive empirical study questioning their ability to measure the diversity and fidelity of the generated graphs. To solve these limitations, they propose a new evaluation metric that computes the Maximum Mean Discrepancy (MMD) between graph representations of the sampled and real graphs, as extracted from an untrained GGM model.

All the reviewers agreed that the research problem is interesting and the overall idea behind the proposed metric is sound and novel. While there were some concerns regarding some details/comparisons/conclusions of the experimental evaluation, the rebuttal managed to cleared up these concerns and all the reviewers eventually supported acceptance.